# Drug-Tolerant *Mycobacterium tuberculosis* Adopt Different Survival Strategies in Alveolar Macrophages of Patients with Pulmonary Tuberculosis

**DOI:** 10.3390/ijms241914942

**Published:** 2023-10-06

**Authors:** Elena G. Ufimtseva, Natalya I. Eremeeva

**Affiliations:** 1Laboratory of Medical Biotechnology, Research Institute of Biochemistry, Federal Research Center of Fundamental and Translational Medicine, 2 Timakova Street, 630117 Novosibirsk, Russia; 2Institute of Disinfectology, F.F. Erisman Federal Scientific Center of Hygiene of the Federal Service on Surveillance for Consumer Rights Protection and Human Well-Being, 18a Nauchniy Proezd, 117246 Moscow, Russia; eremeevani@yandex.ru; 3Scientific Department, Ural Research Institute for Phthisiopulmonology, National Medical Research Center of Tuberculosis and Infectious Diseases of Ministry of Health of the Russian Federation, 50 XXII Partsyezda Street, 620039 Yekaterinburg, Russia

**Keywords:** pulmonary tuberculosis, *Mycobacterium tuberculosis*, alveolar macrophages, drug tolerance, persistence, dormancy, biomarkers, non-acid-fastness, Rv2623

## Abstract

The rapid spread of drug-resistant *M. tuberculosis* (*Mtb*) strains and the phenomenon of phenotypic tolerance to drugs present challenges toward achieving the goal of tuberculosis (TB) elimination worldwide. By using the ex vivo cultures of alveolar macrophages obtained from lung tissues of TB patients after intensive antimicrobial chemotherapy before surgery, different subpopulations of multidrug-tolerant *Mtb* with a spectrum of phenotypic and growth features were identified in the same TB lesions. Our results are indicative of not only passive mechanisms generating nonheritable resistance of *Mtb* to antibiotics, which are associated mainly with a lack of *Mtb* growth, but also some active mechanisms of *Mtb* persistence, such as cell wall and metabolic pathway remodeling. In one of the subpopulations, non-acid-fast *Mtb* have undergone significant reprogramming with the restoration of acid-fastness, lipoarabinomannan expression and replication in host cells of some patients after withdrawal of anti-TB drugs. Our data indicate the universal stress protein Rv2623 as a clinically relevant biomarker of *Mtb* that has lost acid-fastness in human lungs. The studies of *Mtb* survival, persistence, dormancy, and resumption and the identification of biomarkers characterizing these phenomena are very important concerning the development of vaccines and drug regimens with individualized management of patients for overcoming the resistance/tolerance crisis in anti-TB therapy.

## 1. Introduction

Tuberculosis (TB) is caused by the pathogen *Mycobacterium tuberculosis* (*Mtb*) and remains the leading bacterial cause of death in the world [1]. Today, global TB eradication is compromised by an increasingly rapid spread of multidrug-resistant (MDR) and extensively drug-resistant (XDR) TB, which need to be treated with complex combinations of drugs for an extremely prolonged stretch of time, while no cure is often achieved [2,3,4,5]. The pathogen can become tolerant to antibiotics as a result of drug-resistant mutations in the *Mtb* genes that confer resistance, which is both irreversible and heritable [6,7,8]. Another possible cause of TB treatment failure is transient drug tolerance in *Mtb* which is not associated with genomic changes and is defined as phenotypic resistance or persistence [9,10,11,12]. Persistence is the ability of microorganisms to survive prolonged exposure to drugs to which they are fully susceptible, which is thought to be related to the specific physiological states of *Mtb* [13,14,15]. It is believed that the development of persistent *Mtb* in hostile microenvironments, for example, within macrophages or inside granulomatous tissue lesions, is a major barrier to the timely and relapse-free treatment of human TB disease because the current TB drug regimen poorly addresses this pathogen [16,17,18,19]. 

It is assumed that persistent *Mtb* constitutes a reservoir of slow-growing or growth-arrested bacteria, in which many antibiotic target sites are inactive, thereby explaining the ineffectiveness of bactericidal antibiotics [20,21,22]. Several observations correlate the physiological state of the pathogen with dormancy, which is usually associated with the inability of *Mtb* to grow on solid media, its reduced metabolic and growth rates, resulting from alteration of gene expression, and tolerance to antibiotics owing to lack of replication and cell wall remodeling [23,24,25]. 

In laboratory practice, the “dormancy” phenotype of *Mtb* can be induced by different environmental stresses including starvation, hypoxia, oxidative stress, interaction of pathogen with their host, exposure to antibiotics, and various combinations of these [17,26]. Multiple experimental models have been developed to study the persistent/dormant population of *Mtb*, including long-term stationary phase cultures, the Wayne model, the Cornell murine and other animal models, single-cell level studies using microfluidics and time-lapse microscopy, to name a few (reviewed in [16,21,26,27,28]). While the characterization of *Mtb* and other mycobacterial species in these models has helped discover some mechanisms by which persistent *Mtb* emerge and revealed in them the physiological changes that form the basis for their dormancy and multidrug tolerance, no apparent marker of dormant *Mtb* has been identified yet [12,22,25,28,29,30]. Also, it is impossible to determine which of the *Mtb* states in the experimental assays is relevant to human TB disease, where complex interactions between the patient, pathogen, pathological and pharmacological (in the context of antibiotic therapy) factors exist in the specific conditions of the development of *Mtb* infection [31,32]. Defining the cell state of *Mtb* in human TB disease is hindered by the lack of available experimental methods to investigate *Mtb* persistence and dormancy in the patients’ tissues, primarily of the lungs, since *Mtb* is mainly spread through aerosols and engulfed by alveolar macrophages. 

For a deeper understanding of the molecular and cellular biology of *Mtb* and the mechanisms of their survival in the lung tissues of TB patients, we have developed a technique to produce ex vivo cell cultures, mainly of alveolar macrophages, from sections of lung surgically removed from patients with pulmonary TB [33]. As we found in one of our previous works, when replicating bacteria line up along their longitudinal axes in a close parallel arrangement and set themselves into “braids” or “ropes”. *Mtb* cords are revealed in the patients’ alveolar macrophages, and these cords are associated with increased *Mtb* virulence in the guinea pig model of TB disease [34]. We have also found that *Mtb* with different virulence, occurring solitary and as colonies, with or without cording morphology, are exclusively intravacuolar pathogens with intact phagosomal membranes in the viable host cells of TB patients [35]. In a special study, we indicated that the *Mtb*-infected alveolar macrophages in the ex vivo cell cultures were isolated from TB patients’ lung tissues and were not the result of *Mtb* uptake after ex vivo expansion [36]. We identified not only cavity walls, but also tuberculoma walls with insufficient inflammation and excessive fibrosis as being the main niches for *Mtb* survival in alveolar macrophages among the patients’ lung TB lesions examined [37]. While some *Mtb* were found in the colonies, including those with cording morphology, the growth potential of *Mtb* in the alveolar macrophages of TB patients at the time of surgery and ex vivo expansion of cells as well as the mechanisms used by the pathogen to survive antimicrobial therapy remained unknown.

In this study, we used prolonged ex vivo cultures of alveolar macrophages obtained from the lung lesions of patients with pulmonary TB to understand the mechanisms underlying *Mtb* survival in host cells under an intensive antimicrobial chemotherapy before surgery. We have identified persistent *Mtb* infection in the patients’ lungs, where the different subpopulations of multidrug-tolerant *Mtb* that exhibited a spectrum of phenotypic and growth characteristics coexisted in the same lesions. The *Mtb* were able to resume growth quickly, especially when in colonies with cording morphology, in the cells of some patients in a long-term ex vivo culture under antibiotic-free conditions. Shortly after that resumption, non-acid-fast *Mtb* from one of the subpopulations underwent considerable reprogramming with the restoration of their acid-fastness, expression of glycolipid lipoarabinomannan (LAM) in the cell envelope and active replication in the alveolar macrophages of some patients. We have also found *Mtb* expressing the universal stress protein domain-containing protein Rv2623 from the dormancy survival regulator (DosR)-dependent regulon, which had an acid-fast-negative phenotype and none of the virulence factors LAM or 6-kDa early secretory antigenic target (ESAT-6) protein expression in the alveolar macrophages of all patients. Rv2623 protein expression may be used as a clinically relevant biomarker of the *Mtb* that have lost acid-fastness—and so may the *Mtb* dynamics in the prolonged ex vivo cell cultures be as a biomarker of the growth status and physiological state of the pathogen in the host cells of TB patients at the time of surgery and, supposedly, a hallmark of infection relapse after the cessation of antibiotic treatment.

## 2. Results

### 2.1. Nonheritable Persistence of Mtb in Alveolar Macrophages Is the Main Culprit of Treatment Failure in TB Patients

#### 2.1.1. *Mtb*-Infected Alveolar Macrophages Are Determined for all the Patients after an Intensive Anti-TB Chemotherapy

To determine the pathways of *Mtb* survival in the lungs of patients with pulmonary TB, we simultaneously studied *Mtb* infection in alveolar macrophages (1) in the ex vivo cell cultures obtained previously from surgically removed lung TB lesions, such as cavity (cav) wall (patient 6) and tuberculoma (tub) walls (patients 24, 25, and 27–29), and the lung tissues distant from the cavities and tuberculomas (dist or without labeling, ‘distant lung tissue samples’ throughout) for patients 1–5 and 7–29 [33,35,36,37], and (2) on histological sections obtained from the same lung specimens. Patients 4, 15, 23, 28, and 29 had been treated with the first-line anti-TB drugs isoniazid, rifampicin, ethambutol, and pyrazinamide before surgery (Table 1). The treatment of the other patients involved the use of complex combinations of antibiotics, including the first-line anti-TB drugs, especially pyrazinamide, and the second-line anti-TB drugs fluoroquinolones (ofloxacin or levofloxacin), cycloserine, rifabutin, prothionamide, para-aminosalicylic acid, and injectable antibiotics (capreomycin and aminoglycosides, such as amikacin and kanamycin). The acid-fast *Mtb*, as solitary and in colonies, with cording morphology or as irregular clumps, positive for the LAM, ESAT-6, and the cell wall 38-kDa phosphate binding glycoprotein PstS-1 (Ag38) expression, were found in some alveolar macrophages (‘infected alveolar macrophages’ throughout) both after ex vivo culture for 16–18 h and on the histological sections for all the patients studied (Figure 1a–d). The *Mtb* clinical isolates grown on dense Löwenstein–Jensen (L-J) medium were only from the resected lung tissues of patients 6, 8, 10, 11, and 20 (Table 1); however, a substantial number of infected alveolar macrophages (out of the total number of the cells examined, >1% were the host cells with *Mtb*) was determined in the ex vivo cell cultures of patients 3, 5, 9, and 24 (tub) (Figure 1b).

Thus, *Mtb* were able to survive a large number of bactericidal antibiotics that had been given to the patients in a clinical setting for a lengthy time period and had different abilities to grow, i.e., to activate cell division, on a dense medium after surgery.

#### 2.1.2. *Mtb* Are Genetically Susceptible to the Many Antibiotics Used in Clinical Settings

When antimicrobial treatment fails, *Mtb* often become resistant to antibiotics due to mutations in their genes implicated in drug resistance [4,8,38]. Genetic analysis revealed (1) heterogeneity in the presence/absence of mutations to the *Mtb* genes associated with resistance to isoniazid (*katG*, *inhA*), rifampicin (*rpoB*), ethambutol (*embB*), fluoroquinolones (*gyrA*), capreomycin and aminoglycosides (*rrs*, *eis*) and (2) variation in the number of *Mtb* genes affected by these mutations among the patients regardless of the duration of intensive anti-TB chemotherapy before surgery (Table 1). At the same time, although patients 4, 13, and 29 with the wild-type resistance-associated genes in *Mtb* DNA were treated with first-line anti-TB drugs, including the essential antibiotics isoniazid and rifampicin, and the therapy for patients 2, 8, and 14 with the wild-type *rpoB* gene in *Mtb* DNA involved rifampicin or rifabutin in combination with other antibiotics, the alveolar macrophages of these patients were found to be infected with *Mtb* (Figure 1b–d). Also, although the wild-type *gyrA*, *rrs* and *eis* genes were present in *Mtb* DNA of most patients and the treatment regimens included fluoroquinolones for patients 21, 24, 25, and 27, capreomycin/aminoglycosides for patients 2, 3, 5, 7, 11, 12, and 19, and a combination of these antibiotics for patients 1, 16, 22, and 26, these patients’ alveolar macrophages were found to be infected with this pathogen. A similar situation was observed for patients 4–6, 8, 14, 21, 22, and 29 with the wild-type *embB* gene in *Mtb* DNA following treatment with ethambutol. 

Thus, *Mtb* that survived drug attacks in alveolar macrophages are likely to be genetically susceptible to the many antibiotics that were used in the treatment of most patients before surgery.

#### 2.1.3. *Mtb* Survive in Alveolar Macrophages after Exposure to Bactericidal Antibiotics in a Long-Term Ex Vivo Culture

Treatment failure may also be associated with the inability of effective anti-TB drugs to penetrate lung TB lesions and achieve adequate antimicrobial concentrations at infection sites in human tissues, where the pathogen resides [38,39]. This situation prompted us to estimate the efficacy of *Mtb* killing in the ex vivo cell cultures of patients 10, 14, and 15 after three-day exposure to three concentrations of the standard-of-care anti-TB drugs isoniazid (1, 5, and 10 µg/mL), rifampicin (40, 60, and 80 µg/mL), ofloxacin (2, 5, and 10 µg/mL), kanamycin (30, 40, and 50 µg/mL) or capreomycin (30, 40, and 50 µg/mL). The concentrations of these bactericidal antibiotics, which target different pathways of the *Mtb* biogenesis [40], corresponded to the critical, intermediate, and maximum doses used to test *Mtb* tolerance under antibiotic pressure on L–J medium in laboratory settings [41]. Furthermore, we assessed the *Mtb* load in the alveolar macrophages of patients 16–19 exposed only to the critical concentrations of isoniazid (1 µg/mL) and ofloxacin (2 µg/mL) under the same conditions. 

At all concentrations tested, acid-fast *Mtb*, as solitary and in colonies, including those with cording morphology, were revealed in living alveolar macrophages without apoptotic or necrotic morphology for all the patients (Figure 1c and Appendix A), while *Mtb* with heritable resistance to the antibiotics used were found only in patient 10 (Table 1). In our analysis, the number of infected alveolar macrophages often did not differ substantially between the drug-treated cell cultures—even those exposed to the maximum concentrations—and the untreated controls analyzed simultaneously on day 5 (D5) or 6 (D6) after ex vivo expansion of alveolar macrophages. This result aligned with expectations for the isoniazid-treated alveolar macrophages of patients 10, 14–19 and rifampicin-treated alveolar macrophages of patients 10, 15 due to the presence of the resistance-associated mutations in the katG and rpoB genes in *Mtb* DNA, respectively. However, bactericidal activity was low in the alveolar macrophages of patients 14 and 15 after exposure to other antibiotics and in the ex vivo cell cultures of patients 16, 17, and 19 exposed to ofloxacin: although the pathogen was susceptible to these antibiotics (Table 1), it showed phenotypic resistance to them. A notable reduction in the number of infected alveolar macrophages was detected only for patient 18 after exposure to ofloxacin. 

Taken together, our data suggest that although *Mtb* remain genetically susceptible to most anti-TB drugs that were used for intensive antimicrobial therapy before surgery, they develop phenotypic multidrug tolerance or persistence to antibiotic treatment and, therefore, represent the population of bacteria that persist in alveolar macrophages for an extended period of time.

### 2.2. Mtb Resume Active Growth in the Alveolar Macrophages of Some Patients in the Prolonged Ex Vivo Cell Cultures after Withdrawal of Anti-TB Drugs

Growth arrest is regarded as the main hypothesis to explain the persistence of *Mtb*, because the physiological state of non-replication renders bacteria drug insensitive owing to a low target activity [8,13,29]. To assess the growth status of the pathogen (those capable of replicating to be tagged as ‘replicating’, and those incapable of replicating and only surviving in colonies, including those with cording morphology, to be tagged as ‘non-replicating’) at the time of surgery, we analyzed the *Mtb* loads in the patients’ alveolar macrophages stained using the Ziehl–Neelsen (ZN) method after a long-term ex vivo culture under antibiotic-free conditions on days 2, 3 and 4 (D2–D4) and/or on each of days 5–8 (D5–D8) after ex vivo expansion of alveolar macrophages (Figure 1b–d). 

In the prolonged ex vivo cell cultures, we observed a gradual increase in the number of alveolar macrophages with increased *Mtb* loads in them for patients 4, 6 (cav), 7–10, 14, 18, 24 (tub), and 27 (tub) (Figure 1b–d). Also, an increased number of host cells containing upward of 10 *Mtb*, predominantly in colonies with cording morphology or in colonies occurring as irregular clumps, was detected simultaneously with a decrease in the number of alveolar macrophages containing from 3 to 9 *Mtb* in most of these patients (Figure 1d). For the alveolar macrophages of patient 6 (cav), increased replication rates of *Mtb* in colonies with cording morphology resulted in the disruption of some host cells with morphological signs of necrotic cell death (nucleus-free cells, chromatolysis, compromised cytoplasmic membranes and leakage of cell components) and the extracellular formation of large braids or cords with more than 40 acid-fast and LAM-positive *Mtb* in each of them on day 6 (D6) of ex vivo culture (Figure 2a,b). 

Some increase in the number of alveolar macrophages with *Mtb* appearing in the form of colonies, whether as irregular clumps or with cording morphology, was observed in the prolonged ex vivo cell cultures for patients 2, 20, 22, 23, and 26; however, the *Mtb* load in host cells did not increase over time for the other patients, including 24 (dist) and 27 (dist), thereby indicating a very slow division rate and/or the absence of *Mtb* replication in the alveolar macrophages (Figure 1b). Therefore, the *Mtb* colonies, including those with cording morphology for patients 3, 19, 22, and 26, were likely to have been formed in the patients’ host cells long before surgery. This assumption was confirmed by the absence of TB lesions and symptoms of TB disease in the guinea pigs infected with the lung tissue inoculums of patient 19 with a large number of *Mtb* colonies with cording morphology in them [34]. 

Thus, our data indicate that persistent *Mtb* are able to resume active growth in the alveolar macrophages of some patients during the first days in a long-term ex vivo culture under antibiotic-free conditions and then to destroy host cells containing cords with numerous actively replicating *Mtb* in them.

### 2.3. Acid-Fast and LAM-Positive Mtb with an Atypical Morphology Appear in the Alveolar Macrophages of Some Patients in a Long-Term Ex Vivo Culture under Antibiotic-Free Conditions

In the prolonged ex vivo cell cultures of patients 4, 7–9, 18, 20, and 27 (tub), the number of infected alveolar macrophages increased about two- to six-fold from hours 16–18 to D5–D6 of ex vivo culture under antibiotic-free conditions (Figure 1b,c). We also detected some increase in the number of infected alveolar macrophages in the prolonged ex vivo cell cultures of patients 2, 6 (cav), 10, 13, 14, 23, and 24 (tub). To our surprise, a substantial increase in the number of alveolar macrophages with one *Mtb* bacterium in each was revealed in the prolonged ex vivo cell cultures of patients 4, 14, and 24 (tub). Additionally, no change in the number of alveolar macrophages with one *Mtb* bacterium in each was observed for patients 6 (cav), 7, 8, 18 and 27 (tub) after a long-term ex vivo culture (Figure 1d), even though we expected a substantial decrease, because active *Mtb* replication was observed in the alveolar macrophages of these patients. 

Moreover, in the alveolar macrophages of patient 6 (cav), we detected shorter rod-shaped *Mtb* ranging from 0.7 to 2.1 μm in size (the mean being 1.4 ± 0.1 μm, n = 40), on D3 of ex vivo culture, while some earlier, after ex vivo culture for 18 h, only long filamentous *Mtb* ranging from 3.8 to 7.5 μm in size (the mean being 6.6 ± 0.3 μm, n = 40) had been observed (*p* < 0.001, Figure 2a,c,d). We also detected alveolar macrophages with shorter acid-fast or LAM-positive rod-shaped *Mtb* appearing as clusters of two or more bacteria, including those paired in a V-shaped manner, in viable host cells with or without filamentous *Mtb*, whether solitary or in colonies with cording morphology, in them (Figure 2a,c,d). The V-shape of an *Mtb* colony is known to characterize the late stages of *Mtb* cell division [42,43]. For the other patients, we did not notice such striking differences in the shape or size of the *Mtb* re-established in the host cells over the period of observation, because *Mtb* with a different morphology—filamentous/elongated rod-shaped *Mtb* in colonies with cording morphology and shorter rod-shaped *Mtb*, solitary and as irregular clumps—were detected in alveolar macrophages after ex vivo culture for 16–18 h (Figure 1a). 

Given our data, we hypothesized that the re-established acid-fast and LAM-positive *Mtb* that could not be detected earlier in alveolar macrophages by traditional analyses, including the ZN method, based on the unique acid-fastness property of *Mycobacteria* that retain carbolfuchsin dye when decolorized with acid–ethanol, and immunofluorescent staining for virulence factors, could nevertheless re-establish in the prolonged ex vivo cell cultures of some patients after withdrawal of anti-TB drugs and other environmental stresses, such as hypoxia or nutrient deficiency. We called the phenomenon of a sudden appearance of the acid-fast pathogen expressing virulence factors in alveolar macrophages, where it had not been previously detected by traditional methods, “*Mtb* resumption”. 

Thus, analysis of the patients’ alveolar macrophages after a long-term ex vivo culture under antibiotic-free conditions allows us not only to ascertain the *Mtb* growth status and ability to quickly resume division in host cells, but also to detect the resumption of acid-fast and LAM-positive *Mtb* in them. Therefore, *Mtb* dynamics can serve as a biomarker of both active and inactive physiological states of the pathogen in the patients’ lungs at the time of surgery. Additionally, this analysis allows us to make predictions about possible increased risks of *Mtb* infection in patients after cessation of anti-TB chemotherapy.

### 2.4. Mtb within and Outside Alveolar Macrophages of TB Patients Are Devoid of Lipophilic Inclusions

As “… the term dormancy should only apply to bacterial cells and specifically those that are capable of regrowth: it should be a reversible phenomenon…” in [25] (p. 139), we suggested that the re-established *Mtb* could belong to the subpopulation of dormant bacteria that survived exposure to antibiotics and other environmental stresses in a specific phenotypical state in the patients’ alveolar macrophages. Although *Mtb* with lipophilic inclusions in them have been studied only in late stationary phase cultures and some human sputum samples after Auramine–Nile red labeling [44,45], the presence of lipophilic inclusions in *Mtb* is supposed to be a sign of their dormancy phenotype [25,44,45,46]. 

In our work, the lipophilic inclusions in *Mtb* were examined with use of the LAM or ESAT-6 immunofluorescent and Nile red dual-staining for detection of the pathogen and lipids, respectively, in the ex vivo cell cultures of the patients. Although we have previously detected foamy and lipid-rich alveolar macrophages that contained a large amount of lipid droplets in them both in the ex vivo cell cultures and on the histological sections for some patients [35,36], none was detected in the cytoplasm of filamentous or shorter rod-shaped *Mtb* within and outside alveolar macrophages of the patients after ex vivo culture for 16–18 h (Appendix A). The ESAT-6-positive *Mtb* did not produce intracellular lipophilic inclusions in the alveolar macrophages of the *Mtb*-infected guinea pig with clinical signs of TB disease, either (see Appendix A in [35]). On the other hand, we did not observe any Nile red-positive *Mtb* not expressing the virulence factors LAM and ESAT-6. 

Thus, the absence of lipophilic inclusions in the cytoplasm of *Mtb* did not allow us to use this marker as a means to ascertain the dormant state of the pathogen in the patients’ alveolar macrophages. Therefore, other biomarkers are needed. 

### 2.5. Rv2623-Positive Mtb Are Detected Only in Stationary Phase Cultures, but Not in Exponential Phase Cultures of Mtb Beijing Clinical Isolates

The induction of *Mtb* persistence and dormancy is believed to be accompanied by global changes in gene expression due to activation of stress response regulatory mechanisms, including the DosR two-component system with the DosR-dependent regulon composed of more than 50 genes [24,28,47]. Some of these genes, for example, *dosR*, *Rv3130c*, *hspX* (*acr*), *fdxA*, and *narX*, were found to be constitutively overexpressed in *Mtb* clinical isolates belonging to the Beijing genotype family [48]. In our work, the *Mtb* DNA from the resected lung tissues belonged to the Beijing lineage for most patients, excluding patients 3 and 4 [34,35,36]. Therefore, we decided to study the expression of another DosR regulon protein known as the universal stress protein Rv2623, because their *usp* (*rv2623*) gene was among the most highly induced genes of mycobacteria, including *Mtb*, in various experimental models, such as hypoxia [49,50], in vitro infection of mouse and human macrophages [51], and chronical infection in the mouse lungs [52]. As was noted in some models, Rv2623 expression can contribute to the restriction of *Mtb* replication, activation of the dormancy-signaling pathway and the induction of *Mtb* persistence in host cells [53,54]. 

Originally, we analyzed the expression of the *Mtb* markers LAM, Ag38, and Rv2623 in the stationary phase cultures of high- and low-virulence *Mtb* clinical isolates obtained from the resected lung tissues of patients 6, 8, 10, 20 and 11, respectively, all in the Beijing genotype family [33,34]. After 3 months of *Mtb* incubation (D90) on dense L–J medium under standard conditions, we observed a rise in non-acid-fast *Mtb* in bacterial spots obtained from the cultures of all *Mtb* clinical isolates (Figure 3). In an immunofluorescence assay with dual-staining of *Mtb*, we identified four types of expression of the *Mtb* markers in the total population, where some *Mtb* expressed the following markers at once: (1) LAM and Ag38, (2) Ag38 and LAM, with the former prevailing, (3) LAM and Rv2623, with the former prevailing, and (4) Rv2623 alone (Figure 3). These types of *Mtb* marker expression were determined in the stationary phase cultures of all *Mtb* clinical isolates, with LAM/Ag38-positive *Mtb* being prevalent. Rv2623-positive *Mtb* with the normal rod-shaped morphology made up approximately 10% of the bacteria in stationary phase cultures of the *Mtb* clinical isolates examined. 

After analysis of *Mtb* in stationary phase cultures, some *Mtb* from these cultures were replated onto fresh L–J medium under standard conditions. On day 20 (D20) after replating, we revealed only more elongated rod-shaped acid-fast *Mtb*, with LAM and Ag38 expressing in exponential phase cultures, where no Rv2623-positive *Mtb* were observed (Figure 3). The correlation between Rv2623 expression and long-term survival of *Mtb* in stationary phase cultures under nutrient-limited conditions indicated a possible association of this marker with the stress-induced dormancy response of the pathogen.

Thus, we hypothesized that the expression of Rv2623 protein may be used as a marker of the dormant state of *Mtb*, including Beijing family strains, in the alveolar macrophages of TB patients.

### 2.6. Rv2623-Positive Mtb Not Expressing the Virulence Factors LAM and ESAT-6 Are Identified in the Alveolar Macrophages of all the Patients Studied

Using an immunofluorescence assay with dual-staining of *Mtb*, we analyzed the expression of two combinations of *Mtb* markers at once—LAM and Rv2623 or ESAT-6 and Rv2623—in the pathogen in the alveolar macrophages of patients 5 and 10–29 after ex vivo culture for 16–18 h and, in parallel, on the histological sections. Of note, each of the virulence factors LAM and ESAT-6 was targeted using the polyclonal antiserum normally recognizing multiple epitopes of the *Mtb* antigens. As a result, we detected LAM- and ESAT-6-positive *Mtb* not expressing Rv2623 protein and, vice versa, Rv2623-positive *Mtb* not expressing the virulence factors LAM and ESAT-6 in the alveolar macrophages on the cytological and histological preparations and in the caseous center of tuberculomas and small granulomas on the histological sections (Figure 4a and Appendix A). Rv2623-positive rod-shaped *Mtb* were found to occur as solitary and in clusters of two or more bacteria, including V-shaped pairs, but not as colonies with cording morphology, where only LAM- or ESAT-6-positive filamentous *Mtb* were discerned (Figure 4a,c and Appendix A). We did not notice substantial differences in the size of most rod-shaped *Mtb*, solitary or in colonies as irregular clumps, expressing distinct *Mtb* markers, although the Rv2623-positive *Mtb* were overall shorter. We did not observe LAM- or ESAT-6- and Rv2623-positive *Mtb* at once in the same alveolar macrophages in ex vivo culture, while such host cells were identified on the histological sections (for example, patient 24 (tub) in Figure 4a). At the same time, these *Mtb* markers were often detected as occurring together in the same intracellular granules of some alveolar macrophages, along with or without the marker-positive *Mtb* in them, on the histological sections and in the ex vivo cell cultures of the patients (Appendix A).

The number of alveolar macrophages with Rv2623-positive *Mtb* in them ranged from about 10% to almost 50% of host cells between the patients and their lung TB lesions (Appendix A). Furthermore, since the data obtained for the majority of the patients after ex vivo culture for 16–18 h were in good agreement with histological data, we compared only histological data obtained for the patients’ lung specimens divided into four groups (Figure 4b). The compilation of these groups was based on an anatomical and a histological examination of the patients’ lung TB lesions with the assessment of the extent of fibrosis coupled with activation of stress factors (see the table below the graph in Figure 4b), such as the synthesis of the master pro-inflammatory transcriptional regulator nuclear factor-kappa B (NF-*κ*B), the enzyme inducible nitric oxide synthase (iNOS), resulting in the production of nitric oxide, and cyclooxygenase 2 (COX-2), and the production of reactive oxygen species (ROS), which, for most of the patients studied, were assessed in some of our previous works [35,37]. The hostile stresses, especially low oxygen tension and high nitric oxide, are considered to activate the DosR-dependent regulon in *Mtb*, with this activation including the upregulation of Rv2623 protein expression (reviewed in [28]). The lung specimens of the “tuberculomas” group were characterized by excessive fibrosis and insufficient inflammation with a lack of generation of stress factors in the tuberculoma walls of patients 22 and 24–29 (n = 7). For the distant lung TB tissues, the specimens of patients 5, 10, 22, and 23 (n = 4) with extensive fibrosis and the reduced activation of stress factors in alveolar macrophages were included in the “type 1” group. The lung samples of patients 14 and 24–26 (n = 4) with focal fibrosis and patients 15 and 27–29 (n = 4) with minimal signs of fibrosis were assigned to the “type II” and “type III” groups, respectively. In the latter groups, a higher level of pro-inflammatory and microbicidal factors was identified in alveolar macrophages. Although TB lesions of all the groups had different morphological characteristics and hostile environments, they displayed approximately identical Rv2623-positive *Mtb* (solitary or as colonies) loads in the alveolar macrophages, for each group, about a third of host cells were infected (Figure 4b). Changes in the parameters tested between the groups failed to reach statistical significance. Also, some places with Rv2623-positive *Mtb* in them were detected in the caseous center of tuberculomas for patients 22 and 24–26 (n = 4), when half of these places had colonies of Rv2623-positive *Mtb* (Figure 4b and Appendix A). 

Rv2623-positive *Mtb*, solitary or in colonies, excluding those with cording morphology, were also found in the alveolar macrophages of patients 1–4 and 6 (cav) after ex vivo culture for 16–18 h (Figure 4c). At the same time, no lipophilic inclusions were detected in the cytoplasm of Rv2623-positive rod-shaped *Mtb*, solitary or in colonies as irregular clumps, or Rv2623-negative filamentous *Mtb* in colonies with cording morphology on the same cytological preparations for these patients (Figure 4c). 

The examination of the pathogen on the histological sections of the resected lung tissues of patients 32 and 33, who had not been given anti-TB treatment before surgery, revealed Rv2623-positive *Mtb*, solitary and in colonies as irregular clumps, in both alveolar macrophages and the caseum of the tuberculoma walls and, in parallel, the distant lung tissue parts (Appendix A). Rv2623-positive *Mtb* did not express the virulence factors LAM and ESAT-6 or intracellular lipophilic inclusions, either. Of note, Rv2623-positive *Mtb* did not colocalize with filamentous actin in the alveolar macrophages (Appendix A). The numbers of the patients’ alveolar macrophages with Rv2623-positive *Mtb* in them, as well as the *Mtb* loads analyzed on the histological sections by the ZN method, were compared with the data obtained from the histological sections for the other patients after intensive antimicrobial chemotherapy before surgery. 

Thus, Rv2623-positive *Mtb*, mainly belonging to the Beijing genotype family, are revealed in alveolar macrophages and necrotic caseum in the lung tissues of all the patients studied, regardless of the type of TB lesions or environmental niches for *Mtb* survival, and the courses of intensive anti-TB therapy with various sets of antibiotics or without any anti-TB treatment before surgery. No expression of the virulence factors LAM and ESAT-6 is detected in any of the Rv2623-positive *Mtb*.

### 2.7. Rv2623-Positive Mtb Are Acid-Fast-Negative in the Alveolar Macrophages of Patients and Guinea Pig with TB Disease

As is known, the acid-fastness property of *Mtb* is the cornerstone for the diagnosis of TB and identification of the pathogen in the patients’ tissues [55,56]. However, remodeling the cell envelope composition in *Mtb* is expected to lead to alterations in cell wall permeability and to an acid-fast-negative phenotype of the pathogen that is not resistant to decolorization by acid alcohol solutions in the ZN method [55,56]. These characteristics are thought to be associated with the dormant state of *Mtb* during pathogen survival in the experimental models of TB infection (reviewed in [25,57,58,59]) and the lung tissues of TB patients [60,61]. The absence of the main cell wall glycolipid LAM in the Rv2623-positive *Mtb* prompted us to examine the acid-fast property of such *Mtb* by comparing the patients’ infected alveolar macrophages on confocal immunofluorescent images after ex vivo culture for 16–18 h with the same alveolar macrophages re-stained via the ZN method. Previously, we have used this strategy of re-staining cell preparations to establish the unique features of *Mtb* lifestyle in host cells [35]. 

On the ZN re-stained cell preparations of patients 10–29, we did not find any acid-fast *Mtb* in alveolar macrophages with the previously identified Rv2623-positive *Mtb*, solitary and in colonies, in them (Figure 5). Additionally, we revealed a lack of acid-fast staining for Rv2623-positive *Mtb* in alveolar macrophages of the guinea pig with clinical signs of TB disease (Figure 5). The guinea pig was infected in a special study to assess the efficiency of infection control measures in TB hospital departments [35]. The same procedures, first with an immunofluorescence assay of Rv2623-positive *Mtb* and then with ZN re-staining for identification of acid-fast *Mtb*, were applied to cells obtained from the lung tissues of this animal after ex vivo culture for 20 h. Remarkably, we also found the guinea pig’s alveolar macrophage contained both two shorter rod-shaped Rv2623-positive *Mtb*, which were non-acid-fast after ZN re-staining, and acid-fast filamentous *Mtb* that had not expressed the Rv2623 marker in the previous immunofluorescence assay (Figure 5). Therefore, neither Rv2623-positive *Mtb* demonstrated an acid-fastness nor did acid-fast *Mtb* express the Rv2623 protein in the alveolar macrophages. The LAM- and ESAT-6-positive *Mtb* were mainly acid-fast in the patients’ alveolar macrophages after ZN re-staining, while solitary shorter *Mtb* were occasionally non-acid-fast (for example, in patient 27 (tub) in Figure 5). In a mouse model of latent TB infection, we also detected LAM-positive, but non-acid-fast mycobacteria after ZN re-staining in granuloma cells obtained from the spleens of mice infected with the Bacillus Calmette–Guérin (BCG) vaccine (see Figure 5a in [62]).

After ZN re-staining, we did not identify any dead Rv2623-positive *Mtb* in the alveolar macrophages of the patients or the guinea pig: dead bacteria would have stained blue by the dye hematoxylin in our analysis due to an altered cell wall permeability and the destroyed plasma membrane. Therefore, the Rv2623-positive *Mtb* were characterized by an altered cell envelope, but an intact plasma membrane in the alveolar macrophages during TB disease. Therefore, substantial changes in cell wall composition and structure did not compromise the viability of the Rv2623-positive *Mtb* in host cells. Whether the Rv2623-positive *Mtb* replicate in colonies observed in the host cells and quickly restore the biosynthesis of LAM, acid-fastness, and growth in the alveolar macrophages after withdrawal of anti-TB drugs, remains to be known. Overall, our results confirmed the existence of viable *Mtb* with an acid-fast-negative phenotype in the lungs of TB patients and animals and uncovered a significant number of host cells with non-acid-fast *Mtb* in them in various lung TB lesions of the patients studied. This phenomenon is undetectable by ZN staining, which is one of the main methods to detect *Mtb* in the diagnosis of TB infection and disease, and leads to underestimation of *Mtb* burden in the human lungs. 

Thus, our data suggest that the universal stress protein Rv2623 is likely to be a clinically relevant biomarker of viable *Mtb* that had lost their acid-fastness in the patients’ lungs. The identification of this *Mtb* signature has important potential therapeutic implications, including vaccine development, for the treatment and prevention of the human TB disease.

### 2.8. Subpopulations of Phenotypically and Physiologically Distinct Mtb Coexist in the Same Lung Tissues of TB Patients, No Matter What Lesion Type

In our work, we have identified two main pools of *Mtb* in the total population of the pathogen persisting in the alveolar macrophages of TB patients, which were studied with the use of ex vivo cell cultures and, in parallel, histological sections. The *Mtb* were phenotypically different in these pools and were mostly characterized by the presence or absence of the acid-fast property and expression of the virulence factors LAM and ESAT-6 or universal stress protein Rv2623, respectively (Table 2 and Appendix A, Figure 6). Within each pool of *Mtb*, the pathogen survived as a solitary bacterium or in colonies, including those with cording morphology (this applies only to the acid-fast *Mtb*), in host cells and demonstrated various growth potentials in the prolonged ex vivo cell cultures after withdrawal of anti-TB drugs, including the resumption of acid-fast and LAM-positive *Mtb* that replicated in the alveolar macrophages of some patients.

Based on the phenotypic and physiological features of the pathogen, we divided the total *Mtb* population in the alveolar macrophages of the patients into ten subpopulations (Table 2 and Appendix A, Figure 6). The *Mtb* that were capable of actively replicating in a long-term ex vivo culture were assigned to subpopulations 2, 4, 6, and 7, while the *Mtb* with arrested growth were assigned to subpopulations 1, 3, and 5. Normally, we observed the different *Mtb* subpopulations in the patients’ host cells, irrespective of the type of TB lesions (Table 2 and Appendix A). So, the re-established *Mtb* in subpopulation 7 were detected in the alveolar macrophages of many patients (Table 2). The same *Mtb* subpopulations were detected in the host cells obtained both from the cavity wall of patient 6 and the distant lung tissue sample of patient 10. The same *Mtb* subpopulations were revealed in the alveolar macrophages from both the tuberculoma walls and the distant lung tissue samples of patients 25, 28, and 29. However, *Mtb* subpopulations were others in the alveolar macrophages obtained from various lung TB lesions of patients 24 and 27 (Appendix A). 

Overall, the presence of the phenotypically and physiologically different subpopulations of drug-tolerant *Mtb* in the same lung tissues is indicative of not only passive mechanisms generating nonheritable resistance to multiple bactericidal antibiotics, which are associated mainly with a lack of *Mtb* growth in host cells, but also some active mechanisms of *Mtb* persistence, such as cell wall and metabolic pathway remodeling, which together help the pathogen survive anti-TB chemotherapy and other environmental stresses in infected alveolar macrophages.

## 3. Discussion

Our data indicate that phenotypically multidrug-tolerant *Mtb* not acquiring genetic mutations for resistance to the antibiotics used adopt diverse survival strategies in the alveolar macrophages of patients with active TB disease. The pathogen is able to survive antimicrobial chemotherapy in various subpopulations that are characterized by different phenotypic and physiologic features and coexist in various lung TB lesions. The characterization of the pathogen growth status identified acid-fast *Mtb* both in the growth-arrested state and with the ability to active division, as well as the re-established *Mtb* with the restoration of their acid-fastness, LAM synthesis, and replication, in the prolonged ex vivo cultures of the patients’ alveolar macrophages after withdrawal of anti-TB drugs. Additionally, Rv2623-positive *Mtb* having an acid-fast-negative phenotype and not expressing the virulence factors LAM and ESAT-6 were revealed for each TB patient studied. Taken together, our findings demonstrate a complex composition of *Mtb* infection in the lungs of TB patients, where the pathogen can dynamically change under the modified environmental conditions. 

We could not determine whether the Rv2623-positive non-acid-fast *Mtb* quickly transformed into the LAM-positive and acid-fast *Mtb* or the re-established *Mtb* belonged to other subpopulations that were not detected in our work. However, the behavior of the *Mtb* re-established in the prolonged ex vivo cultures of the patients’ alveolar macrophages after withdrawal of anti-TB drugs resembled the behavior of the pathogen re-established after termination of anti-TB treatment in the Cornell murine model of *Mtb* persistence and dormancy [63,64], while it took this phenomenon several weeks to take shape in mice and only some days in the ex vivo cell cultures of TB patients. Note that *Mtb* were unculturable for both mice in the Cornell model and most TB patients after antimicrobial chemotherapy, although the *Mtb* DNA was detected in the PCR-analysis in all studies [33,65]. On the whole, further analysis of the phenomenon of *Mtb* resumption in the alveolar macrophages of TB patients should include the identification of the mechanisms of its development and the question of involvement of Rv2623-positive *Mtb*. 

Interestingly, both the regrowth and resumption of the pathogen in the alveolar macrophages of some patients were achieved without the use of any special resuscitation-promoting factors or other products of *Mtb* in a long-term ex vivo culture. Meanwhile, many substances were necessary for resuscitation of *Mtb* in the different in vitro dormancy models with a wide spectrum of stressful conditions in stationary phase [66,67,68,69] and from the sputum of patients with active TB disease [70], while the *Mtb* transition from a dormant, nonreplicating state to an actively dividing state was estimated by a standard colony-forming assay. We hypothesize that *Mtb*, as an intravacuolar pathogen in the viable host cells of TB patients [35], are likely to be able to monitor stressful environments, including exposure to anti-TB drugs, through the interaction of the *Mtb* phagosomes and human endocytic pathways, as we have established previously [35], and to quickly resume the growth in favorable conditions, especially in colonies with cording morphology, for some patients. It is likely that some Rv2623-positive *Mtb* reside within phagosomes in the patients’ alveolar macrophages and interact with the host endosomal system for monitoring environmental conditions. These assumptions, however, require further investigation. 

It remains in question whether the phenotype of the Rv2623-positive non-acid-fast *Mtb* is a reliable indication of the dormant state of the pathogen in the lungs of TB patients, because higher levels of the Rv2623 protein in cerebrospinal fluid have been proposed as a diagnostic biomarker of not only active, but also latent TB meningitis in humans [71]. Additionally, the Rv2623 protein was identified as a negative regulator in the cell envelope lipid biosynthesis [54] with, as expected, a decrease in LAM during the entry of *Mtb* into non-growth persistence and dormancy [72]. In our work, the morphological signs of Rv2623-positive *Mtb* were compared with the morphology of acid-fast and LAM- or ESAT-6-positive rod-shaped *Mtb* detected in the lungs of the patients, and were found to be significantly different from the characteristics of coccoid *Mtb* isolated in some in vitro models of *Mtb* dormancy [67,73,74] and from the infected animal tissues and the cavitary walls of TB patients [75]. At the same time, the signatures of *Mtb* clinical isolates on dense L-J medium in our study were similar to the characteristics of the model object *M. smegmatis* in stationary phase cultures and after exposure to different stress signals, which were also characterized by a loss of LAM expression and acid-fastness, but rapidly regained these properties after replating onto fresh medium [55,76,77]. Taken together, these results strongly suggest that the modulation of the LAM content that was detected in both the pathogenic and non-pathogenic mycobacteria, is a well-coordinated, growth phase- and environmental-signal-regulated process likely affecting the cell wall integrity and unique acid-fast property of mycobacteria. Overall, while in vitro and in vivo animal models remain some of the main sources of knowledge about the mechanisms of *Mtb* survival under multiple stressful conditions (reviewed in [68,69]), they are unlikely to reflect the actual situation in human TB disease or the true interaction between the pathogen and host cells under the treatment of TB patients. Therefore, future studies of *Mtb* persistence, dormancy, and resumption as well as the identification of biomarkers characterizing these phenomena and the biological states and growth features of the pathogen in humans are very important. 

In our study, we have identified several biomarkers that characterize *Mtb* persistence in the alveolar macrophages of TB patients. First, *Mtb* dynamics in the prolonged ex vivo cell cultures after withdrawal of anti-TB drugs may represent a biomarker of the growth status and the physiological state of the pathogen, whether solitary or in colonies, including those with cording morphology, in the patients’ host cells at the time of surgery. Second, our analysis allows potential reactivation of *Mtb* infection to be predicted in patients after cessation of antibiotic therapy and enables more accurate inferences to be made about the presence of higher-virulence *Mtb* than our previous suggestion, that is analysis of *Mtb* cording formation in the alveolar macrophages of TB patients [34]. In fact, initiating the regrowth of persistent/dormant *Mtb* followed by exposure to anti-TB drugs as one of the anti-TB treatment approaches (reviewed in [13,78]) can quickly eliminate drug-tolerant cells of the pathogen, as we detected in the prolonged ex vivo cell culture of patient 18 after exposure to ofloxacin. However, this strategy is extremely unsafe, because it is difficult to control the activity of this process in the patients’ host cells, as we observed in the prolonged ex vivo cell cultures of patient 6.

We have previously noted [79], a sudden emergence of fibroblast-like cells in prolonged ex vivo cell cultures may indicate the presence of mesenchymal stem cells in the lung TB lesions, which were detected for the tuberculoma wall of patient 27, but not for any other patients or TB lesions. Interestingly, we did not reveal any *Mtb* in these cells, although mesenchymal stem cells are expected to be a reservoir of *Mtb* infection in human TB disease and thus help the pathogen survive the anti-TB drug experience [80]. Therefore, it is necessary to proceed further with analysis of human host cells with the use of the opportunities provided by the ex vivo cell cultures obtained from different lung tissues of TB patients. 

Third, the Rv2623 protein may be used as a clinically relevant biomarker of *Mtb* that had lost their acid-fastness in the patients’ lungs. We also assume that the Rv2623-positive and non-acid-fast state of *Mtb* is common for some pathogens that have survived in both human TB lesions and animal lungs and is not related to anti-TB drug exposure alone. These assumptions, however, need to be investigated. In addition, other mechanisms can lead to a loss of acid-fastness in *Mtb*, for example, those associated with defects in the biosynthesis of mycolic acids in the pathogen during TB infection of mice [81,82].

Overall, a deeper understanding of the molecular and cellular biology of persistent *Mtb* in the lungs of TB patients, including transition of the pathogen from a dormant nonreplicating state into an actively dividing state, is critical to improve the management of the disease and design more effective drugs or drug regimen to eradicate *Mtb* infection worldwide.

## 4. Materials and Methods

### 4.1. Patients and Lung Tissue Samples

Lung tissue specimens were obtained from 29 patients with clinically active pulmonary TB at the Department of Thoracic Surgery of the Ural Research Institute for Phthisiopulmonology (Yekaterinburg, Russia) affiliated with the National Medical Research Center of Tuberculosis and Infectious Diseases of the Ministry of Health of the Russian Federation (Moscow, Russia) over the period from August 2014 to July 2018 as described in [33,35,36,37]. The patients’ nomenclature used is explained in [33]. These patients were residents of the Ural province of Russian Federation and had received TB treatment under the supervision of medical staff at their local clinics. These patients had been characterized in detail (age, treatment, attendant diseases, surgery and others) previously in Table 1 and Appendix A in [33,35,36,37]. Lung tissue samples were also obtained from patients 32 (59 years old, male, without attendant diseases, surgery—upper lobe of right lung) and 33 (55 years old, female, without attendant diseases, surgery—upper lobe of right lung) with pulmonary TB at the Department of Thoracic Surgery of the Novosibirsk Regional Clinical Oncology Dispensary (Novosibirsk, Russia) over the period from November 2022 to March 2023. The diagnoses of patients 32 and 33 were verified by the pathomorphological studies of the lung lesions after surgery. Patients 32 and 33 were residents of the Siberia province and the Sakha Republic (Yakutia) of Russian Federation, respectively, and had not received any TB treatment before surgery. All patients gave written informed consent for collection of clinical correlates, tissue collection, research testing under the Ethics Committees of the Ural Research Institute for Phthisiopulmonology of the National Medical Research Center of Tuberculosis and Infectious Diseases- and Novosibirsk Regional Clinical Oncology Dispensary-approved protocols (27/2014/07/02 and 15/2021/11/16). Patient studies were conducted according to the Declaration of Helsinki. All the patients studied had tuberculomas and other fibrotic and caseotic TB lesions in the lungs and had been referred for the surgical management of pulmonary TB. Seven patients (6–10, 22, and 23) had cavities in the lungs. Immediately after surgery, pieces of lung tissue (~0.5–30 g) obtained from lung parts about 5 cm away from macroscopic TB lesions (tuberculomas and cavities) were collected for patients 1–5, 7–29, 32, and 33, pieces of the tuberculoma walls were collected for patients 24, 25, 27–29, 32, and 33, and only the cavity wall was collected for patient 6.

### 4.2. Analysis of Drug-Resistance Mutations in Mtb Genes

For isolation of mycobacterial DNA, the lung tissue samples obtained from the surgically resected lung parts of patients 1–29 were incubated for 60 min in 0.5 mL of Amplitube-Prep solution for *Mtb* inactivation (Sintol, Moscow, Russia), and then an Amplitube-PB kit (Sintol, Russia) was used for DNA extraction. Mutations associated with resistance to isoniazid (*katG*, *inhA*), rifampicin (*rpoB*), ethambutol (*embB*), fluoroquinolones (*gyrA*, *gyrB*), capreomycin and aminoglycosides (*rrs*, *eis*) were analyzed using a microarray TB-TEST assay (BIOCHIP-IMB, Moscow, Russia) according to the manufacturer’s instructions.

### 4.3. Bacterial Strains and Cultures

The highly and low virulent *Mtb* clinical isolates 14–329, 14–319, 15–169, 15–446, and 15–218 were obtained from the lung tissue homogenates of patients 6, 8, 10, 20, and 11, respectively, with pulmonary tuberculosis (TB) during our earlier work [33,34] and used in this study. The *Mtb* clinical strains were cultured on dense L-J medium (Himedia Laboratories, India) at +37 °C under standard conditions. The aliquots of the *Mtb* strains from each stationary and, after replating of *Mtb* onto fresh L–J medium, exponential phase culture were added to an uncoated glass microscope slide, dried, fixed with 10% formaldehyde solution in PBS for 10 min at room temperature.

### 4.4. Ex Vivo Isolation and Culture of Human Cells

Alveolar macrophages from the specimens of surgically resected lung lesions of patients 1–29 were produced as described in [33,35,36,37]. In brief, samples of lung tissue were cut into small pieces and, for separating cell suspension containing alveolar macrophages from closed granulomatous fibrotic tissue, were further rubbed through a metal screen of a sieve with pores 0.5–2.0 mm in diameter in phosphate-buffer saline (PBS, pH 7.4). Cell pellets were centrifuged at 400 g for 5 min at room temperature and placed to 24-well plates (Orange Scientific, Belgium) with cover glasses (~8 × 8 mm in size) in the bottom and cultured for 16–18 h in 0.5 mL of RPMI 1640 complete growth medium containing 10% fetal bovine serum, 2 mM glutamine and 50 µg/mL gentamicin (BioloT, Russian Federation) at +37 °C in an atmosphere containing 5% CO_2_. At hours 16–18 of ex vivo culture, after removal of growth medium with dead cell debris, monolayer cultures of human cells on the cover slips were washed twice with PBS for removal of nonadherent cells. At this time point, more than 90% of cells obtained from the cavity and tuberculoma walls and distant parts of resected lung tissue of all studied patients were found to be alveolar macrophages [33,35,36,37]. Besides alveolar macrophages, five more cell types were observed: dendritic cells, neutrophils, lymphocytes, fibroblasts, and multinucleate Langhans giant cells, but the population sizes of these cells were much lower (see Appendix A and Table 1 in [33,36]). Further, some ex vivo cell cultures of the patients were cultured in 0.5 mL of the complete growth medium without any antibiotics for 2–8 days at +37 °C in an atmosphere containing 5% CO_2_. Some ex vivo cell cultures of the patients were then exposed to different concentrations (see text) of anti-TB drugs: isoniazid (Sigma-Aldrich, St. Louis, MO, USA, I3377), rifampicin (Nanjing Pars Biochem, Nanjing, China, R6056), ofloxacin (Sigma-Aldrich, USA, O8757), kanamycin (Sigma-Aldrich, USA, K4000), or capreomycin (Nanjing Pars Biochem, China, MB1047),—in the complete growth medium for 3 days at +37 °C in an atmosphere containing 5% CO_2_.

### 4.5. Guinea Pig Infection and Ex Vivo Isolation of Animal Cells

Experiments involving animals were performed in accordance with “The Guidelines for Manipulations with Experimental Animals” issued by the Russian Ministry of Health (guideline 755) and European convention for the protection of vertebrate animals used for experimental and other scientific purposes (ETS no. 123, Strasbourg) and were approved by the Ethical Committee of the Ural Research Institute for Phthisiopulmonology of the National Medical Research Center of Tuberculosis and Infectious Diseases (Yekaterinburg, Russia, 27/2014/07/02). Female outbred guinea pig (10 weeks old, 320 g) was obtained from the Science and Production Association for Medical and Immunological Preparations (Mikrogen, Bashkortostan, Russia) and maintained under standard vivarium conditions, with water and food provided ad libitum. This guinea pig was infected with 0.5 mL of washout from the hands of the nurse in PBS into the right inguinal fold as described in [35] to determine the degree of *Mtb* contamination in the TB hospital departments and to evaluate the efficiency of infection control measures. After infection, guinea pig was closely monitored, paying attention to clinical and morphologic sings of TB disease. At day 79 post infection, the guinea pig with clinical signs of TB disease were euthanized and examined macroscopically and histologically for changes and TB lesions in the right lung, liver and spleen. Also, the lung piece of this guinea pig was homogenized in PBS and plated on L–J medium as described in [34]. After 4 weeks of incubation at +37 °C with obligatory visual control on a weekly basis, mycobacterial colonies were identified and confirmed as being *Mtb* using standard procedures. Cells were isolated from the left lung of this guinea pig following the same procedure as for the alveolar macrophages from the resected lungs of the TB patients and then cultured in 0.5 mL of the complete growth medium under the same conditions as human cells. At hour 20 of ex vivo culture, only alveolar macrophages (~80% of the cell population) and polymorphonuclear neutrophils (~20% of the cell population) were observed in the guinea pig cell cultures. 

### 4.6. Cell Staining

At hours 16–20 of ex vivo culture, after removal of growth medium with dead cell debris, some monolayer cultures of human and guinea pig cells on cover slips were washed with PBS and fixed with 4% formaldehyde solution in PBS for 10 min at room temperature. To visualize acid-fast *Mtb* within host cells, after washing with PBS, some cell preparations were stained via the ZN method. After ZN staining, the cells were further counterstained with Mayer’s hematoxylin. The other cell cultures were used for staining with antibodies and other fluorescent reagents. 

In an immunofluorescence assay using dyes and/or antibodies, the cell preparations were incubated with 10 µM of Nile red dye (Invitrogen, Waltham, MA, USA, N1142) for 15 min at +37 °C in 5% CO_2_ before fixation. Next, the cell preparations were fixed as described above, washed with PBS, blocked in PBS solution containing 2% BSA, and finally, incubated first with primary antibodies, then with secondary antibodies. Some of the fixed cell preparations were permeabilized with 0.3% Triton X-100 solution in PBS for 2 min before blocking. Primary antibodies were against *Mycobacteria* LAM (Abcam, England, ab20832, 1:200 dilution), *Mtb* ESAT-6 (courtesy of E.V. Deineko, Federal Research Center Institute of Cytology and Genetics, SB RAS, Novosibirsk, Russia, 1:300 dilution), Rv2623 (Abcam, England, ab24291, 1:1000 dilution), and Ag38 (Abcam, England, ab183165, 1:1000 dilution. Fluorescent visualization of markers was enabled using secondary goat polyclonal DyLight 488- and DyLight 594-conjugated anti-rabbit IgG (Thermo Fisher Scientific, Waltham, MA, USA, 35553, and 35561, respectively, 1:400 dilution), Alexa 488- and Alexa 555-conjugated anti-mouse IgG (Thermo Fisher Scientific, USA, A11001, and Invitrogen, USA, A21422, respectively, 1:400 dilution) antibodies. The cell preparations were incubated with the appropriate antibodies for 60 min at room temperature. Fluorescent staining was analyzed using the VECTASHIELD Mounting Medium with DAPI (4′,6-diamidino-2-phenylindole) (Vector Laboratories, Newark, CA, USA, H-1200).

Some of the fixed spots with stationary or exponential phase cultures of *Mtb* clinical isolates on glass microscope slides were stained via the ZN method. Other *Mtb* preparations were incubated with the appropriate primary antibodies to *Mycobacteria* LAM and *Mtb* Ag38 or Rv2623 as described above.

The patients’ and guinea pig’s alveolar macrophages stained for LAM, ESAT-6, and Rv2623 were analyzed all along the cells’ height via confocal microscopy and, after washing from VECTASHIELD^®^ Mounting Medium in PBS for 20 min, re-stained for acid-fast *Mtb* by the ZN method as described above. Then, alveolar macrophages with acid-fast *Mtb* were viewed using an Axioskop 2 *plus* light microscope (Zeiss, Jena, Germany). Finally, host cells on the confocal fluorescent images were compared with alveolar macrophages on the ZN images. 

### 4.7. Histology

The histological sections of the resected lung tissues of TB patients were prepared as described in [37]. In brief, the resected lung parts of the TB patients were cut into pieces. One portion of lung pieces was collected for producing alveolar macrophages as described above. The other portion of lung pieces was fixed with 4% formaldehyde solution in PBS (pH 7.4) for 20 h at +4 °C. After fixation, the lung tissues were washed with PBS, incubated with 30% sucrose in PBS (pH 7.4) for 20 h at +4 °C, frozen in Tissue-Tek O.C.T. Compound (Sakura Finetek, Torrance, CA, USA, 4583) at −25 °C, and sectioned at 16-µm slides on a Microtome Cryostat HM550 (Microm, Walldorf, Germany) at the Shared Center for Microscopic Analysis of Biological Objects of the Institute of Cytology and Genetics, SB RAS (Novosibirsk, Russian Federation). Sections were air-dried on SuperFrost Plus slides (Thermo Fisher Scientific, USA) and stained by the ZN and immunofluorescent methods as described above. Primary rabbit monoclonal primary antibodies to human CD14 (Spring Bioscience, Pleasanton, CA, USA, M492, 1:100 dilution), Alexa 488- and TRITC-labeled phalloidin dyes (Thermo Fisher Scientific, USA, A12379, and Sigma-Aldrich, USA, P1951, respectively, 1:100 dilution) were used, too. Some of the histological sections were treated within 45 min in 0.3% Triton-X100 solution. All the histological sections were incubated with the appropriate primary antibodies for 20 h at +4 °C and with the appropriate secondary antibodies for 60 min at room temperature. Fluorescent staining was analyzed using the ProLong Gold Antifade Mountant with DAPI (Thermo Fisher Scientific, USA, P36935). 

### 4.8. Microscopy

The histological sections, cytological and other preparations were examined at the Shared Center for Microscopic Analysis of Biological Objects of the Institute of Cytology and Genetics, SB RAS (Novosibirsk, Russian Federation), using an Axioskop 2 *plus* microscope (Zeiss) and objectives with various magnifications (Zeiss), and photographed using an AxioCam HRc camera (Zeiss); the images were analyzed using the AxioVision 4.7 microscopy software (Zeiss). Cell preparations stained with fluorescent dyes were examined under an LSM 780 laser scanning confocal microscope (Zeiss) using the LSM Image Browser and ZEN 2010 software (Zeiss). The human cells and *Mtb* in host cells were counted separately on each preparation for each patient in each test. More than 1000 alveolar macrophages were analyzed at each cytological preparation for each patient. The size of *Mtb* was analyzed using the ImageJ 1.53t software. For the histological preparations, three un-serial tissue sections from each individual sample were analyzed for each patient. 

### 4.9. Statistical Analyses

Statistical data processing was performed using Prism 6.0 (GraphPad Software) and Microsoft Excel 2010 with each statistical test, definitions of mean and standard error of the mean (SEM), and the number of samples per group (n, referring to individual cells or the lung tissues of TB patients) indicated in the corresponding text, figure legends, and panels. Statistical significance for the comparisons between the datasets was determined using Student’s *t*-test. Differences were considered statistically significant at *p* ˂ 0.05.

## 5. Conclusions

Our data indicate the presence of multiple forms of *Mtb* infection with various microbial compositions and pathological signs for each TB patient studied. Drug-tolerant *Mtb* that persist in alveolar macrophages of TB patients under an intensive antimicrobial chemotherapy before surgery exhibit a spectrum of phenotypic and physiologic features and coexist in various lung TB lesions. Both passive mechanisms associated mainly with a lack of *Mtb* growth and some active mechanisms of *Mtb* persistence, such as cell wall and metabolic pathway remodeling, generate nonheritable resistance of *Mtb* to bactericidal antibiotics. The pathogen can dynamically change with the restoration of its acid-fastness, LAM expression, and active division in the prolonged ex vivo cell cultures of some patients after withdrawal of anti-TB drugs. Rv2623-positive *Mtb* exhibiting an acid-fast-negative phenotype and not expressing the virulence factors LAM and ESAT-6 are revealed for all TB patients. Rv2623 protein expression is proposed as a clinically relevant biomarker of viable *Mtb* that have lost acid-fastness in human lungs. Moreover, the *Mtb* dynamics in the prolonged ex vivo cell cultures may be used as a biomarker of the growth status and physiological state of the pathogen in the host cells of TB patients at the time of surgery and, supposedly, a hallmark of infection relapse after the cessation of antibiotic treatment. In conclusion, our findings demonstrate a complex composition of *Mtb* infection in the lungs of TB patients and provide a new insight into the pathogenesis of *Mtb* during human TB disease. A better understanding of the mechanisms and pathways of pathogen survival, persistence, dormancy, and resumption in human lungs is key to developing vaccines and drug regimens with individualized management of TB patients for overcoming the resistance/tolerance crisis in the treatment of *Mtb* infection toward global TB eradication.

## Figures and Tables

**Figure 1 ijms-24-14942-f001:**
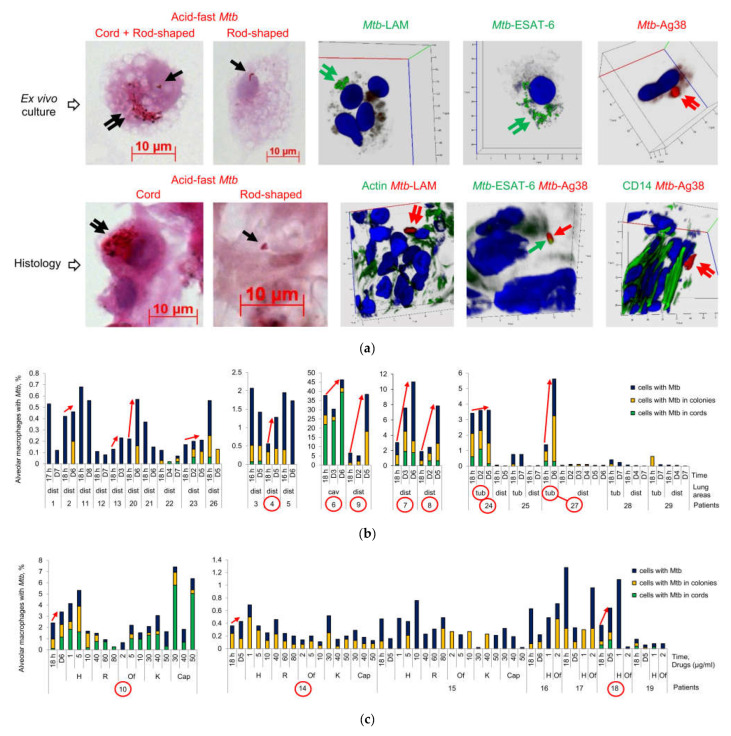
*M. tuberculosis* (*Mtb*) survive a long-term antimicrobial chemotherapy in TB patients and resume active growth in alveolar macrophages of some patients in the prolonged ex vivo cell cultures after withdrawal of anti-TB drugs. (**a**) *Mtb* are acid-fast after Ziehl–Neelsen (ZN) staining and produce the virulence factors LAM, ESAT-6, and Ag38 stained with appropriate specific antibodies (green or red signals) in alveolar macrophages after ex vivo culture for 18 h and on histological sections obtained from the resected lung tuberculoma wall of patient 24 after treatment with multiple antibiotics before surgery. Black, green, and red arrows point to acid-fast, LAM- or ESAT-6-positive, and LAM- or Ag38-positive *Mtb*, respectively, on representative cytochemical and confocal 3D immunofluorescent images. Double arrows point to filamentous *Mtb* in colonies with cording morphology. Nuclei are stained by DAPI (blue signal). Collagen fibers are strongly autofluorescent (green signal) on histological immunofluorescent images. The scale bars are 10 μm each; (**b**–**d**) *Mtb* loads are estimated in the ex vivo cultures of alveolar macrophages obtained from the cavity (cav) or tuberculoma (tub) walls and the tissues distant (dist or without labeling) from the macro-TB lesions of the patients’ resected lung parts; (**b**,**c**) The number of alveolar macrophages with any *Mtb* (solitary or as colonies, including those with cording morphology), with the colonies of *Mtb*, including the cords, and with *Mtb* only in cord-colonies in them, all expressed as the percentage of the total number of alveolar macrophages stained by the ZN method, represents the *Mtb* load in host cells of patients after ex vivo culture for 16–18 h (16–18 h) and from two to eight days (D2–D8) without treatment with anti-TB drugs, and (**c**) after three days of ex vivo exposure to different concentrations of antibiotics, such as isoniazid (H), rifampicin (R), ofloxacin (Of), kanamycin (K), or capreomycin (Cap). Red arrows point to a trend towards increase in the number of *Mtb*-infected alveolar macrophages in the prolonged ex vivo cell cultures at D2–D8 versus 18 h. The numbers of patients, whose prolonged ex vivo cell cultures were characterized by this trend, are marked by red ovals; (**c**) Exposure of the patients’ alveolar macrophages to different concentrations of antibiotics does not eliminate drug-resistant or drug-susceptible *Mtb* in host cells in ex vivo culture during three days of treatment with anti-TB drugs; (**d**) Quantification of the alveolar macrophages with a particular number of *Mtb* in them expressed as the percentage of the total number of the *Mtb*-infected alveolar macrophages done for the patients, whose numbers are marked by red ovals in (**b**,**c**), after ex vivo culture for 18 h and D2–D6. Red and blue arrows point to trends towards, respectively, an increase and a decrease in the number of alveolar macrophages with a particular number of *Mtb* in them after ex vivo culture for various time points.

**Figure 2 ijms-24-14942-f002:**
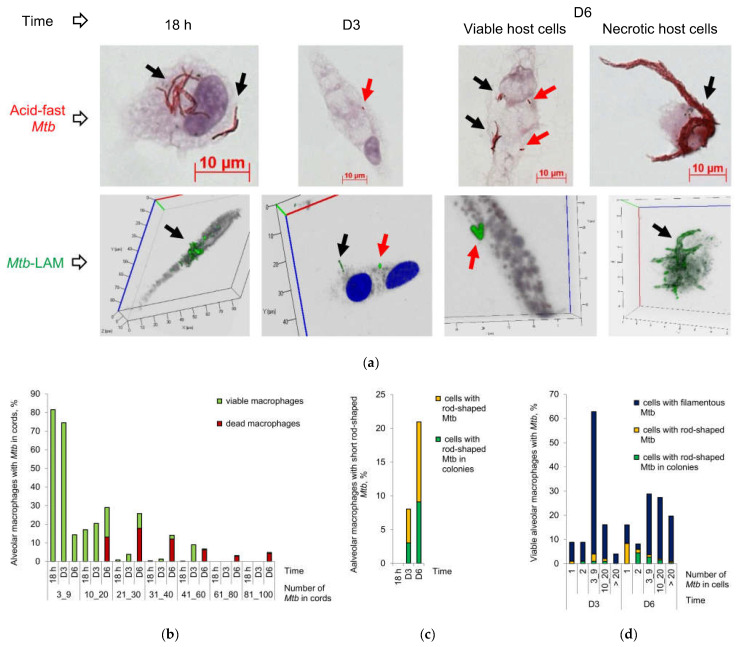
*M. tuberculosis* (*Mtb*) with a distinct morphology re-establish in alveolar macrophages obtained from the cavity wall of patient 6 in the prolonged ex vivo cell cultures after withdrawal of anti-TB drugs. (**a**) Representative cytochemical and confocal 3D immunofluorescent images demonstrate *Mtb* with a distinct morphology, which are acid-fast after ZN staining and express LAM, stained with specific antibodies (green signal), in the patient’ alveolar macrophages after ex vivo culture at different time points. Single green and red arrows point to *Mtb* with a long filamentous and a shorter rod-shaped morphology, respectively. Double black and green arrows point to colonies with cording morphology and filamentous *Mtb* in them. Nuclei are stained by DAPI (blue signal). The scale bars are 10 μm each; (**b**) The number of viable and dead alveolar macrophages with different numbers of *Mtb* in colonies with cording morphology in them is expressed as the percentage of the total number of infected alveolar macrophages after ex vivo culture at all time points; (**c**) The number of alveolar macrophages with any short rod-shaped *Mtb* (solitary or as colonies) and with the colonies of *Mtb* with a shorter rod-shaped morphology are expressed as the percentage of the total number of the patient’ alveolar macrophages with acid-fast *Mtb* (filamentous or as rod-shaped) in them after ex vivo culture at all time points; (**d**) The number of viable alveolar macrophages with a particular number of filamentous *Mtb* (solitary or as colonies), with short rod-shaped *Mtb* (solitary or as colonies), and with the colonies of short rod-shaped *Mtb* in them expressed as the percentage of the total number of the patient’ alveolar macrophages with a particular number of acid-fast *Mtb* (filamentous or as rod-shaped) in them after a long-term ex vivo culture for D3 and D6.

**Figure 3 ijms-24-14942-f003:**
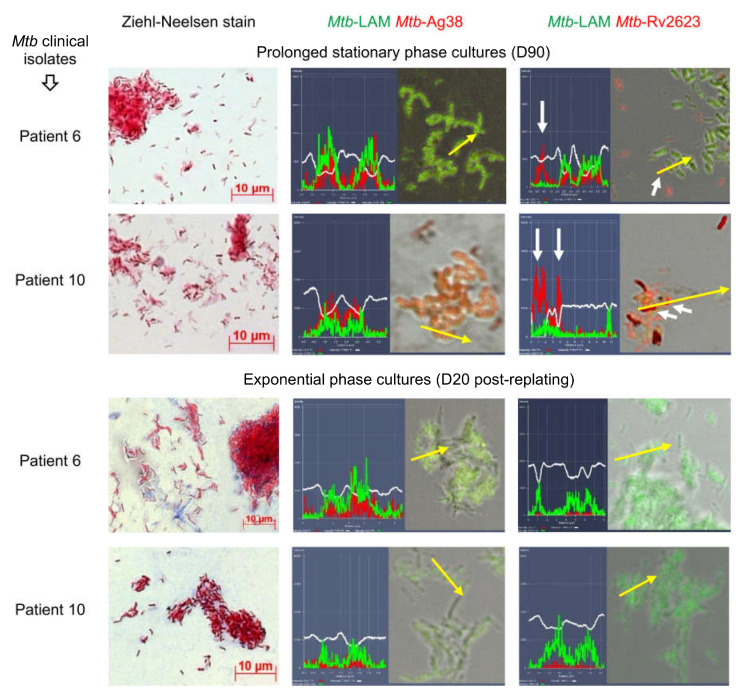
*M. tuberculosis* (*Mtb*) clinical isolates obtained from the lung tissues of patients 6 and 10 produce the universal stress protein Rv2623 only in stationary phase (D90), but not in exponential phase (D20 after replating onto fresh L–J medium) cultures on dense L–J medium. Representative cytochemical images of *Mtb* after ZN staining and the merged confocal immunofluorescent and phase contrast images of *Mtb* stained with appropriate specific antibodies (green and red signals) are demonstrated. White short and long arrows on the merged images and profile graphs, respectively, point to the same Rv2623-positive *Mtb* with no LAM production in them. Yellow arrows on the merged images point to the areas for constructing profile graphs. The scale bars are 10 μm each.

**Figure 4 ijms-24-14942-f004:**
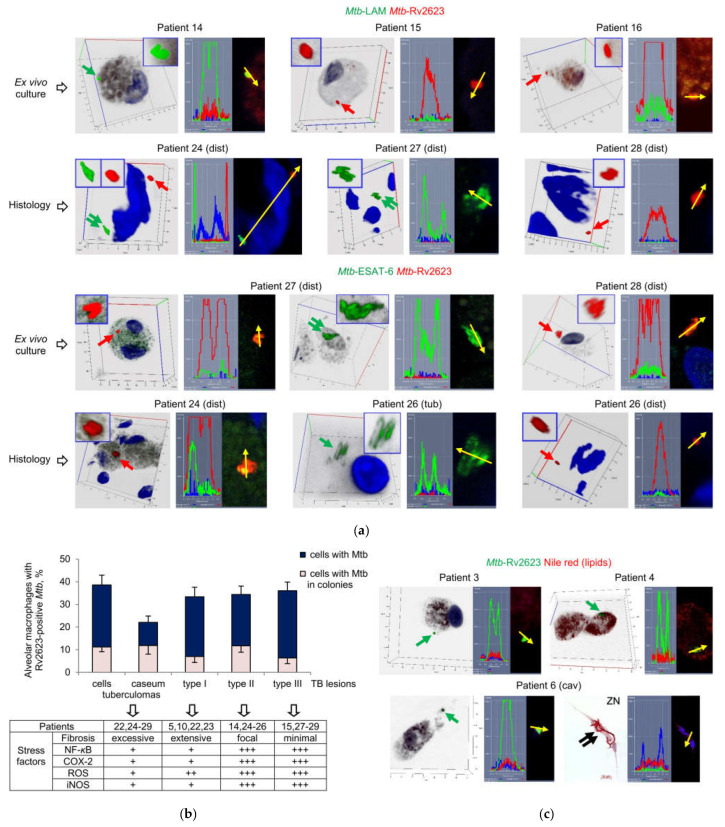
Rv2623-positive *Mtb* not expressing the virulence factors LAM and ESAT-6 are identified in the alveolar macrophages of TB patients. (**a**) Representative confocal 3D immunofluorescent images stained simultaneously with appropriate specific antibodies demonstrate LAM- or ESAT-6-positive *Mtb* (green signal) not expressing the Rv2623 protein and, vice versa, Rv2623-positive *Mtb* (red signal) not expressing the virulence factors LAM and ESAT-6 in the patients’ alveolar macrophages on the histological sections and after ex vivo culture for 16–18 h. Close-ups of the parts of the images with *Mtb*, solitary and in colonies, including those with cording morphology, are shown in the upper left or right corners. Green and red arrows point to the LAM- or ESAT-6- and Rv2623-positive *Mtb*, respectively; (**a**,**c**) *Mtb* are characterized in alveolar macrophages obtained from the cavity (cav) and tuberculoma (tub) walls and the tissues distant (dist or without labeling) from the macro-TB lesions of the patients’ resected lung parts. To the right of the 3D immunofluorescent images and (**c**) the image after ZN re-staining: profile images of the arrow-marked *Mtb* are shown. Yellow arrows point to the areas for constructing profile graphs. Nuclei are stained by DAPI (blue signal). Double arrows point to *Mtb* in colonies with cording morphology; (**b**) The number of alveolar macrophages or caseous regions with any Rv2623-positive *Mtb* (single or as colonies) and with the colonies of Rv2623-positive *Mtb*, both expressed as the percentage of the total number of host cells or caseous regions with LAM- or ESAT-6- and Rv2623-positive *Mtb* in them on the histological sections. The total data of two dual-staining on the LAM/Rv2623- and ESAT-6/Rv2623-markers of *Mtb* for each patient are used for the construction of the graph. Mean ± SEM. Below the graph, the table presents four groups of patients with different TB lesions and characteristics, including the extent of fibrosis and activation of stress factors. Symbols (+), (++), and (+++) indicate activation of stress factors in a single, rare, and the majority of the alveolar macrophages examined; (**c**) Representative confocal 3D immunofluorescent images demonstrate the absence of intracellular lipophilic inclusions stained with Nile red dye (red signal), in the Rv2623-positive *Mtb* stained with specific antibodies (green signal), in the patients’ alveolar macrophages after ex vivo culture for 16–18 h. Green and black arrows point to Rv2623-positive *Mtb* and acid-fast *Mtb* in colonies with cording morphology, respectively. The scale bar is 10 μm.

**Figure 5 ijms-24-14942-f005:**
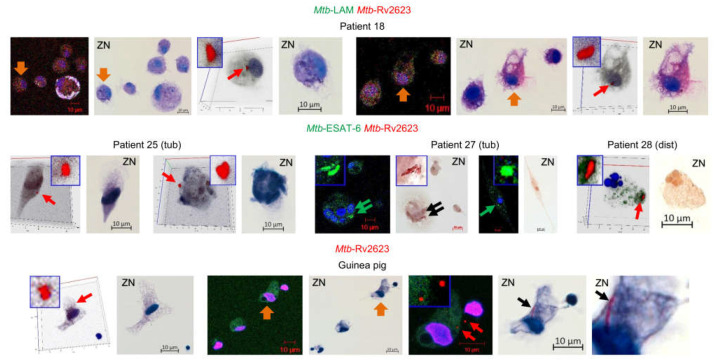
Rv2623-positive *M. tuberculosis* (*Mtb*) reveal an acid-fast-negative phenotype in alveolar macrophages of patients and guinea pig with TB disease. Representative confocal 3D or single immunofluorescent images of host cells stained with antibodies reacting with the *Mtb* LAM or ESAT-6 (green signal) and *Mtb* Rv2623 (red signal) markers, and images of the same host cells after ZN re-staining (ZN) demonstrate Rv2623-positive *Mtb*, solitary and as colonies, with an acid-fast-negative phenotype in the patients’ and guinea pig’s alveolar macrophages after ex vivo culture for 18–20 h. Nuclei are stained by DAPI (blue signal). Close-ups of the parts of the images with *Mtb*, solitary and in colonies, including those with cording morphology, are shown in the upper left or right corners. Green and red arrows point to the LAM- or ESAT-6- and Rv2623-positive *Mtb*, respectively. Brown short arrows point to the host cells with the Rv2623-positive *Mtb* in them. Black arrows point to the acid-fast *Mtb* on the ZN images. Double arrows point to *Mtb* in colonies with cording morphology *Mtb* are characterized in the ex vivo cultures of alveolar macrophages and, in parallel, on the histological sections obtained from the cavity (cav) and tuberculoma (tub) walls and the tissues distant (dist or without labeling) from the macro-TB lesions of the patients’ resected lung parts. The scale bars are 10 μm each.

**Figure 6 ijms-24-14942-f006:**
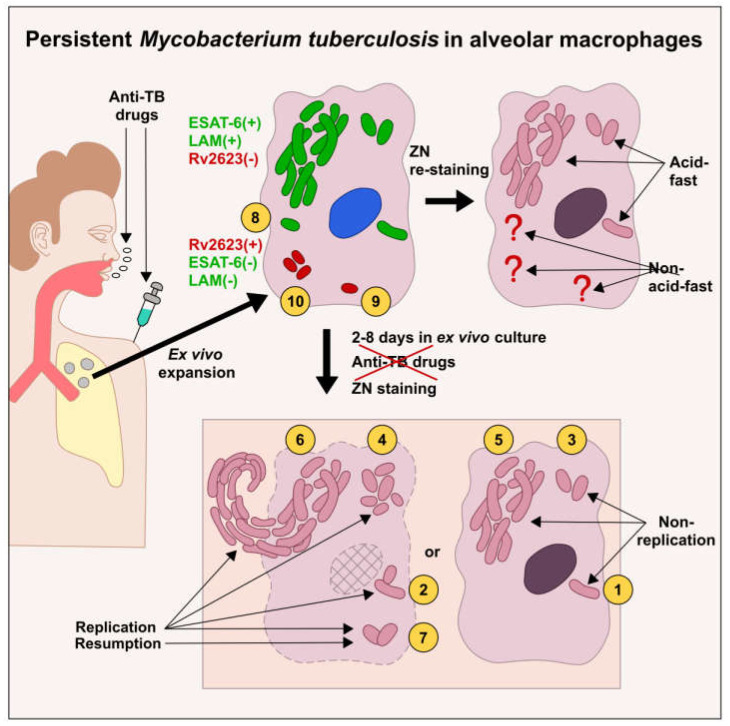
Graphical summary presents the different subpopulations of multidrug-tolerant *Mtb* with a spectrum of phenotypic and growth features in alveolar macrophages of TB patients. The numbers of *Mtb* subpopulations from Table 2 are shown in yellow circles. (?), the absence of acid-fast *Mtb* after ZN re-staining.

**Table 1 ijms-24-14942-t001:** Drug-resistance mutations in the *M. tuberculosis* (*Mtb*) genes for the patients with pulmonary TB included in the study.

Patients	Mutations in the *Mtb* Genes Associated with Drug Resistance to							Diagnosis	Treatment ^1^		Virulence ^2^	*Mtb* Isolate ^3^
	H		R	E	Fq	AK			Drugs	Mo nths		
	*katG*	*inhA*	*rpoB*	*embB*	*gyrA*	*rrs*	*eis*					
1	S315T1	wt	S531L	wt	wt	wt	wt	MDR TB	Z Pt Cap Pas Cs Of	24	n.d.	−
2	S315T1	wt	wt	wt	wt	wt	wt	TB	H R Z Pt Cap	9	n.d.	−
3	S315T1	wt	S531L	wt	wt	wt	wt	MDR TB	H Z Pt Cap Pas Cs	9	n.d.	−
4	wt	wt	wt	wt	wt	wt	wt	TB	H R Z E	5	n.d.	−
5	S315T1	wt	S531L	wt	wt	wt	wt	MDR TB	H R Z E Pt Cap Cs	11	L	−
6	S315T1	wt	S531L	wt	wt	a1401g	wt	MDR TB	Z E Pt Cap Cs	155	H	+
7	S315T1	wt	S531L	M306I	wt	wt	wt	MDR TB	H Z E Pt Cap Cs	5	n.d.	−
8	S315T1	wt	wt	wt	wt	wt	wt	TB	H Z E Pt Rfd	5	H	+
9	S315T1	wt	S531L	wt	wt	g1484t	wt	MDR TB	Z Rfd Cap Pas Cs	31	n.d.	−
10	S315T1	wt	S531L	M306V	D94G	a1401g	wt	XDR TB	Z E Pt Cap Pas Cs	41	H	+
11	S315T1	wt	S531L	wt	wt	wt	wt	MDR TB	Z Pt Cap Pas Cs	74	L	+
12	S315T1	wt	S531L	wt	wt	wt	wt	MDR TB	Z Pt Cap Pas Cs	19	L	−
13	wt	wt	wt	wt	wt	wt	wt	TB	H R Z Pt Pas	3	L	−
14	S315T1	wt	wt	wt	wt	wt	wt	TB	R Z E Pas	6	L	−
15	S315T1	wt	S531L	M306V	wt	wt	wt	MDR TB	H R Z E	58	L	−
16	S315T1	c15t	S531L	M306V	wt	wt	wt	MDR TB	Z Pt Cap Pas Cs Of	40	L	−
17	S315T1	wt	S531L	M306V	wt	wt	wt	MDR TB	H Rfd Z E	5	L	−
18	S315T1	wt	S531L	M306V	wt	wt	wt	MDR TB	H R Z E	27	IM	−
19	S315T1	wt	S531L	wt	wt	wt	wt	MDR TB	Z Pt Cap Pas Cs	26	L	−
20	S315T1	wt	S531L	M306V	wt	wt	wt	MDR TB	H Z E Rfd	113	H	+
21	wt	wt	wt	wt	wt	wt	wt	TB	Z E Pas Cs Of	14	L	−
22	S315T1	wt	H526N	wt	wt	wt	wt	MDR TB	Z E Pas Cs K Of	5	L	−
23	S315T1	t8g	S531L	D354A	wt	wt	c12t	MDR TB	H R Z E	12	L	−
24	S315T1	wt	S531L	G406A	wt	wt	c12t	MDR TB	Z Pt Cap Pas Cs Lfx	7	L	−
25	S315T1	wt	S531L	M306V	wt	wt	g10a	MDR TB	Z Pt Cap Pas Lfx	9	L	−
26	S315T1	wt	S531L	G406A	wt	wt	wt	MDR TB	Z Pt Pas Cs K Lfx	9	L	−
27	S315T1	wt	S531L	M306V	wt	wt	c14t	MDR TB	E Am Cs Lfx	6	L	−
28	ND	ND	ND	ND	ND	ND	ND	TB	H R Z E	3	L	−
29	wt	wt	wt	wt	wt	wt	wt	TB	H R Z E	9	L	−

ND, not determined; n.d., not done; wt, wild type; (+), is present; (−), is absent. ^1^ before surgery. Drugs: AK, aminoglycosides; Am, amikacin; Cap, capreomycin; Cs, cycloserine; E, ethambutol; Fq, fluoroquinolones; H, isoniazid; K, kanamycin; Lfx, levofloxacin; Of, ofloxacin; Pas, para-aminosalicylic acid; Pt, protionamide; R, rifampicin; Z, pyrazinamide. ^2^ was characterized in the guinea pig model of TB disease [34]. Degree: L, low; IM, intermediate; H, high. ^3^ was grown on L-J medium [33].

**Table 2 ijms-24-14942-t002:** Features of the *M. tuberculosis* (*Mtb*) that survived antibiotic therapy in patients with pulmonary TB.

*Mtb* Phenotypes				SP	TB Patients
Acid-Fastness	Markers	Survival State ^1^	Growth Status ^2^		
+	+LAM +ESAT-6 −Rv2623	solitary	non-replicating	1	all the patients and lung TB lesions studied
			replicating	2	2–4, 6 (cav), 7–10, 17, 18, 20, 22, 23, 24 (tub), 27 (tub)
		in colonies (as irregular clumps)	non-replicating	3	3–5, 6 (cav), 7–10, 14–19, 22, 23, 24 (tub and dist), 25 (tub and dist), 26, 27–29 (tub and dist)
			replicating	4	4, 6 (cav), 7–10, 16–18, 22, 23, 24 (tub), 27 (tub)
		in colonies with cording morphology	non-replicating	5	3, 19, 22, 26
			replicating	6	6 (cav), 7, 8, 10, 18, 24 (tub), 27 (tub)
		re-established ^2^	replicating	7	4, 6 (cav), 7–10, 13, 14, 18, 20, 23, 24 (tub), 27 (tub)
		solitary	ns	8	14–16, 18, 24 (tub), 27 (tub)
−	−LAM −ESAT-6 +Rv2623	solitary	ns	9	all the patients and lung TB lesions studied
		in colonies (as irregular clumps)	ns	10	3, 4, 6 (cav), 10, 14–16, 18,19, 22, 23, 24 (tub and dist), 25 (tub and dist), 26, 27–29 (tub and dist)

SP, subpopulations; (+), is present; (−), is absent; ns, not studied. The phenotype of *Mtb* was determined in alveolar macrophages obtained from the cavity (cav) and tuberculoma (tub) walls and the tissues distant (dist or without labeling) from the macro-TB lesions of the patients’ resected lung parts after ^1^ ex vivo culture for 16–18 h and in ^2^ the prolonged ex vivo cell cultures after withdrawal of anti-TB drugs.

## Data Availability

The datasets generated for this study are available on request to the corresponding author.

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
