# Peer review of "Drug-Tolerant Mycobacterium tuberculosis Adopt Different Survival Strategies in Alveolar Macrophages of Patients with Pulmonary Tuberculosis"

_ijms, 2023, doi:10.3390/ijms241914942_

Round 1

Reviewer 1 Report

Elena G. et al.,  have developed a  technique to produce ex vivo cell cultures, mainly of alveolar macrophages, from the  lung parts surgically removed from patients with pulmonary TB. They used prolonged ex vivo cultures of alveolar macrophages from the lung lesions of patients with pulmonary TB to understand the mechanisms underlying Mtb survival in host cells.

The study is well-designed and presented. However, some concept are missing.

1)      How does the main finding of the study affect TB patients?

2)      Does it influence the medicine taking in the patients? How?

3)      According to some data, envelope composition of drug-resistant isolates can hypothesize that drug-susceptible and drug-resistant strains which will adapt differently to the lung microenvironments, driving different infection outcomes. Did they find such results in their study? Please discuss this issue as well. Did the authors assess cell envelope composition of drug-resistant M.tb strains and their metabolic responses and adaptation?

4)      It is better to summarize methods through flowchart in order to ease the reading according to massive data.

5)      The abstract and conclusion must improve and carry the main massage of the study highlighted.

6)      In many studies, dendritic cells have been found as the key immune cells to assess cellular pathogens and have been introduced as vaccine candidates for future. Pleas add such this possibility regarding TB in discussion for further work if sound possible ( https://doi.org/10.1016/j.virusres.2021.198403)

Author Response

Response to Reviewer #1

(IJMS-2587999
Drug-tolerant Mycobacterium tuberculosis adopt different survival strategies in alveolar macrophages of patients with pulmonary tuberculosis)

I would like to thank the Reviewer for careful reading of our manuscript and for his/her comments that led to improving of article’s contents.

Comment 1:

“1) How does the main finding of the study affect TB patients?”

Answer for Comment 1:

So far, it does not affect the treatment of TB patients in any way. In the Russian Federation, the treatment of tuberculosis is regulated by Federal Law No. 77-FZ of 18.06.2001 (as amended on 05.12.2022) "On prevention of the spread of tuberculosis in the Russian Federation" and the Order of the Ministry of Health of the Russian Federation of November 15, 2012 No. 932n "On approval of the procedure for providing medical care to tuberculosis patients". Consequently, many studies of the mechanisms of Mtb survival in the lungs of TB patients are necessary to adjust the regulations for the treatment of TB patients and make changes to the Orders of the Ministry of Health of the Russian Federation.

Comment 2:

“2) Does it influence the medicine taking in the patients? How?”

Answer for Comment 2:

Currently, it does not affect the medicine taking in TB patients in any way, since their treatment is regulated by Federal Law No. 77-FZ and the Order of the Ministry of Health of the Russian Federation No. 932n (see above). To change them, it is necessary to conduct a lot of studies on the mechanisms of the relationship between Mtb and host cells in human lungs with the identification of the pathways and mechanisms used by the pathogen for survival bactericidal anti-TB drugs. In this study, we have presented some part of our work in this direction and it will continue further.    

Comment 3:

“3) According to some data, envelope composition of drug-resistant isolates can hypothesize that drug-susceptible and drug-resistant strains which will adapt differently to the lung microenvironments, driving different infection outcomes. Did they find such results in their study? Please discuss this issue as well. Did the authors assess cell envelope composition of drug-resistant M.tb strains and their metabolic responses and adaptation?”

Answer for Comment 3:

Now, in Results, a passage in lines 554-565 reads:

“As is known, the acid-fastness property of Mtb is the cornerstone for the diagnosis of TB and identification of the pathogen in the patients’ tissues [55,56]. However, remodeling the cell envelope composition in Mtb is expected to lead to alterations in cell wall permeability and to an acid-fast-negative phenotype of the pathogen that is not resistant to decolorization by acid alcohol solutions in the ZN method [55,56]. These characteristics are thought to be associated with the dormant state of Mtb during pathogen survival in the experimental models of TB infection (reviewed in [25,57-59]) and the lung tissues of TB patients [60,61]. The absence of the main cell wall glycolipid LAM in the Rv2623-positive Mtb prompted us to examine the acid-fast property of such Mtb by comparing the patients’ infected alveolar macrophages on confocal immunofluorescent images after ex vivo culture for 16-18 hours with the same alveolar macrophages re-stained by the ZN method.”

In the manuscript (Discussion), a passage in lines 723-739 reads:

“Additionally, the Rv2623 protein was identified as a negative regulator in the cell envelope lipid biosynthesis [54] with, as expected, a decrease in LAM during the entry of Mtb into non-growth persistence and dormancy [72]… At the same time, the signatures of Mtb clinical isolates on dense L-J medium in our study were similar to the characteristics of the model object M. smegmatis in stationary phase cultures and after exposure to different stress signals, which were also characterized by a loss of LAM expression and acid-fastness, but rapidly regained these properties after replating onto fresh medium [55,76,77]. Taken together, these results strongly suggest that the modulation of the LAM content that was detected in both the pathogenic and non-pathogenic mycobacteria, is a well-coordinated, growth phase- and environmental signal-regulated process likely affecting the cell wall integrity and unique acid-fast property of mycobacteria.”

In our work, we studied only the expression of the major Mtb cell wall component glycolipid lipoarabinomannan (LAM) in the cell envelope of the Rv2623-positive and -negative Mtb survived anti-TB drugs in the patients’ alveolar macrophages. In our manuscript, we used many references on original articles and reviews of other scientists about the changes in the Mtb cell wall (see references [54-61,72,76,77]), but did not discuss their interesting results and our experimental data in more detail, since the characteristics of other proteins, glycolipids, and lipids in the cell envelope composition of drug-tolerant Mtb are the subject for our future research.

Comment 4:

“4) It is better to summarize methods through flowchart in order to ease the reading according to massive data.”

Answer for Comment 4:

I agree with the Reviewer that the Materials and Methods is some long. I shortened the Materials and Methods in order to ease the reading. I did not add a flowchart so as not to increase the number of figures.

Comment 5:

“5) The abstract and conclusion must improve and carry the main massage of the study highlighted.”

Answer for Comment 5:

I agree with the Reviewer and tried to improve the abstract and conclusion somewhat.

Now, in Abstract, a passage in lines 14-29 reads:

“The rapid spread of drug-resistant M. tuberculosis (Mtb) strains and phenomenon of phenotypic tolerance to drugs challenge the goal of tuberculosis (TB) elimination worldwide. By using the ex vivo cultures of alveolar macrophages obtained from lung tissues of TB patients after intensive antimicrobial chemotherapy before surgery, different subpopulations of multidrug-tolerant Mtb with a spectrum of phenotypic and growth features were identified in the same TB lesions. Our results indicative of not only passive mechanisms generating nonheritable resistance of Mtb to antibiotics, which are associated mainly with a lack of Mtb growth, but also some active mechanisms of Mtb persistence, such as cell wall and metabolic pathway remodeling. In one of the subpopulations, non-acid-fast Mtb have undergone significant reprogramming with the restoration of acid-fastness, lipoarabinomannan expression and replication in host cells of some patients after withdrawal of anti-TB drugs. Our data indicate the universal stress protein Rv2623 as a clinically relevant biomarker of Mtb that have lost acid-fastness in human lungs. The studies of Mtb survival, persistence, dormancy, and resumption and the identification of biomarkers characterizing these phenomena are very important to develop vaccines and drug regimens with the individualized management of patients for overcoming the resistance/tolerance crisis in anti-TB therapy.”

Now, in Conclusions, a passage in lines 972-995 reads:

“Our data indicate the presence of multiple forms of Mtb infection with various microbial compositions and pathological signs for each TB patient studied. Drug-tolerant Mtb that persist in alveolar macrophages of TB patients under an intensive antimicrobial chemotherapy before surgery exhibit a spectrum of phenotypic and physiologic features and coexist in various lung TB lesions. Not only passive mechanisms that are associated mainly with a lack of Mtb growth, but also some active mechanisms of Mtb persistence, such as cell wall and metabolic pathway remodeling, generate nonheritable resistance of Mtb to bactericidal antibiotics. The pathogen can dynamically change with the restoration of their acid-fastness, LAM expression, and active division in the prolonged ex vivo cell cultures of some patients after withdrawal of anti-TB drugs. Rv2623-positive Mtb having an acid-fast-negative phenotype and not expressing the virulence factors LAM and ESAT-6 are revealed for all TB patients. Rv2623 protein expression is proposed as a clinically relevant biomarker of viable Mtb that have lost acid-fastness in human lungs. Also, the Mtb dynamics in the prolonged ex vivo cell cultures may be used as a biomarker of the growth status and physiological state of the pathogen in the host cells of TB patients at the time of surgery and, supposedly, a hallmark of infection relapse after the cessation of the antibiotic treatment. In conclusion, our findings demonstrate a complex composition of Mtb infection in the lungs of TB patients and provide a new insight into the pathogenesis of Mtb during human TB disease. A better understanding of the mechanisms and pathways of pathogen survival, persistence, dormancy, and resumption in human lungs is the key to develop vaccines and drug regimens with the individualized management of TB patients for overcoming the resistance/tolerance crisis in the treatment of Mtb infection toward global TB eradication.” 

 Comment 6:

“6) In many studies, dendritic cells have been found as the key immune cells to assess cellular pathogens and have been introduced as vaccine candidates for future. Please add such this possibility regarding TB in discussion for further work if sound possible (https://doi.org/10.1016/j.virusres.2021.198403).”

Answer for Comment 6:

In the manuscript (Materials and Methods), a passage in lines 849-854 reads:

“In this time point, more than 90% of cells obtained from the cavity and tuberculoma walls and distant parts of resected lung tissue of all studied patients were found to be alveolar macrophages that had a large number of denser dark inclusions in the cytoplasm and were what is called smokers’ macrophages [33,35-37]. Besides alveolar macrophages, five more cell types were observed: dendritic cells, neutrophils, lymphocytes, fibroblasts, and multinucleate Langhans giant cells, but the population sizes of these cells were much lower (see Tables S3 and 1 in [33,36]).”

In our work with different lung lesions of TB patients, we very closely monitored the presence of dendritic cells in these lung areas for all the patients studied (see also Table S3 in [33] and Table 1 in [36]). In general, the cells obtained from different lung lesions in the ex vivo cell cultures as well as the immune cells examined on the histological sections were found to be alveolar macrophages. Mtb were largely found in alveolar macrophages and rarely neutrophils and Langhans giant cells, and only in one dendritic cell, while one of our previous observations was that many dendritic cells, including those with bacteria in them, with the ex vivo monolayer cell cultures migrate from individual granulomas obtained from the spleens of mice after one month and two months of infection with the Bacillus Calmette–Guérin (BCG) vaccine [E. Ufimtseva, “Investigation of Functional Activity of Cells in Granulomatous Inflammatory Lesions from Mice with Latent Tuberculous Infection in the New Ex Vivo Model,” Clinical and Developmental Immunology, 2013, Volume 2013, Article ID 371249: 1-14. https://doi.org/10.1155/2013/371249; E. Ufimtseva, “Mycobacterium-Host Cell Relationships in Granulomatous Lesions in a Mouse Model of Latent Tuberculous Infection,” BioMed Research International, 2015, Volume 2015, Article ID 948131: 1-16. doi: https://doi.org/10.1155/2015/948131], as well as for mice in the study mentioned by the Reviewer. Therefore, in this manuscript, we are able to study the mechanisms of Mtb survival only in alveolar macrophages. Of course, the total absence of dendritic cells in the lung tissues of patients with active TB disease is a problem for effective treatment of Mtb infection. This problem requires a separate study and a very detailed discussion of its results in the future manuscript.  

Once again, I would like to thank the Reviewer for careful reading of our manuscript and for their comments that led to improving of article’s contents.

Elena Ufimtseva,             

PhD,

Research Institute of Biochemistry, Federal Research Center of Fundamental and Translational Medicine, Novosibirsk, Siberia, Russia.

Reviewer 2 Report

General Comments:

The paper titled "Drug-Tolerant Mycobacterium tuberculosis Survival Strategies in Alveolar Macrophages of Pulmonary Tuberculosis Patients" by Elena G. Ufimtseva and Natalya I. Eremeeva presents an intriguing investigation into the survival mechanisms adopted by drug-tolerant Mycobacterium tuberculosis (Mtb) within alveolar macrophages of patients with pulmonary tuberculosis. The study addresses an important aspect of tuberculosis research, shedding light on the diverse strategies Mtb employs for persistence. Overall, the research contributes valuable insights to the field. Below are my specific comments and suggestions for consideration:

Strengths:

Relevance and Significance: The paper addresses a critical issue in the field of tuberculosis research by focusing on the mechanisms of drug-tolerant Mtb survival. The investigation into the varying strategies adopted by Mtb in alveolar macrophages of tuberculosis patients is highly relevant and provides valuable information for combating drug resistance.

Clear Objectives and Methods: The study's objectives are well-defined, and the methods are adequately described to allow for replication. The use of ex vivo cultures from patients' lung tissues enhances the clinical relevance of the findings.

Innovative Findings: The paper reveals intriguing findings, particularly the identification of different subpopulations of drug-tolerant Mtb with distinct survival strategies. The reprogramming of non-acid-fast Mtb and the identification of the universal stress protein Rv2623 as a potential biomarker add novelty to the study.

Suggestions for Improvement:

Introduction : The introduction could benefit from a clearer and more focused presentation of the research's importance. Highlight the research gap and clearly state the objectives of the study to engage readers effectively.

Clarity in Terminology: Certain terms such as "non-acid-fastness," "acid-fast-negative phenotype," and "biomarkers" might be unfamiliar to all readers. Consider providing concise explanations or including a glossary to aid understanding.

Discussion Enrichment: The discussion section could be expanded to provide a more comprehensive interpretation of the results. Address the broader implications of the findings in the context of tuberculosis treatment and potential clinical applications.

Minor Revisions:

In the abstract, consider rephrasing "emergence of drug-resistant strains of M. tuberculosis (Mtb)" for clarity.

Throughout the paper, ensure consistency in formatting references and maintain adherence to the journal's guidelines.

Results: excellent

Discussion: add data about predictors factor of adverse event (see and cite  Predictors for Pulmonary Tuberculosis Outcome and Adverse Events in an Italian Referral Hospital: A Nine-Year Retrospective Study (2013-2021). Ann Glob Health. 2022 Apr 26;88(1):26. doi: 10.5334/aogh.3677. )

Conclusion:

The paper titled "Drug-Tolerant Mycobacterium tuberculosis Survival Strategies in Alveolar Macrophages of Pulmonary Tuberculosis Patients" presents an important investigation into the survival mechanisms of drug-tolerant Mtb within host cells. The study's novel findings and insights contribute significantly to the understanding of tuberculosis persistence and drug resistance. Addressing the suggestions provided would enhance the clarity and impact of the paper. Overall, the research conducted is commendable and will undoubtedly be of interest to the tuberculosis research community.

Rating:

High-quality work with substantial contributions. Requires minor revisions for clarity and completeness.

Confidentiality Note:

I confirm that I have reviewed this paper confidentially and do not have any competing interests that could affect my review

minor English linguage are need

Author Response

Response to Reviewer #2

(IJMS-2587999
Drug-tolerant Mycobacterium tuberculosis adopt different survival strategies in alveolar macrophages of patients with pulmonary tuberculosis)

I would like to thank the Reviewer for careful reading of our manuscript and for his/her comments that led to improving of article’s contents.

Comment 1:

“Introduction: The introduction could benefit from a clearer and more focused presentation of the research's importance. Highlight the research gap and clearly state the objectives of the study to engage readers effectively.”

Answer for Comment 1:

Now, in Introduction, a passage in lines 92-99 reads:

“While some Mtb were found in the colonies, including those with cording morphology, the growth potential of Mtb in the alveolar macrophages of TB patients at the time of surgery and ex vivo expansion of cells as well as the mechanisms used by the pathogen to survive antimicrobial therapy remained unknown.

In this study, we used prolonged ex vivo cultures of alveolar macrophages obtained from the lung lesions of patients with pulmonary TB to understand the mechanisms underlying Mtb survival in host cells under an intensive antimicrobial chemotherapy before surgery.”

Comment 2:

“Clarity in Terminology: Certain terms such as "non-acid-fastness," "acid-fast-negative phenotype," and "biomarkers" might be unfamiliar to all readers. Consider providing concise explanations or including a glossary to aid understanding.”

Answer for Comment 2:

Now, in the Results, a passage in lines 344-348 reads:

“Given our data, we hypothesized that the re-established acid-fast and LAM-positive Mtb that could not be detected earlier in alveolar macrophages by traditional analyses, including the ZN method, based on the unique acid-fastness property of Mycobacteria that retain carbolfuchsin dye when decolorized with acid-ethanol, and immunofluorescent staining for virulence factors, could nevertheless re-establish in the…”

Now, in the Results, a passage in lines 554-558 reads:

“As is known, the acid-fastness property of Mtb is the cornerstone for the diagnosis of TB and identification of the pathogen in the patients’ tissues [55,56]. However, remodeling the cell envelope composition in Mtb is expected to lead to alterations in cell wall permeability and to an acid-fast-negative phenotype of the pathogen that is not resistant to decolorization by acid alcohol solutions in the ZN method [55,56]. These…”

Now, in Discussion, a passage in lines 743-746 reads:

“Therefore, future studies of Mtb persistence, dormancy, and resumption as well as the identification of biomarkers characterizing these phenomena and the biological states and growth features of the pathogen in humans are very important.”

Comment 3:

“Discussion Enrichment: The discussion section could be expanded to provide a more comprehensive interpretation of the results. Address the broader implications of the findings in the context of tuberculosis treatment and potential clinical applications.”

Answer for Comment 3:

Now, in all clinics of the Russian Federation, the treatment of patients with TB is regulated by Federal Law No. 77-FZ of 18.06.2001 (as amended on 05.12.2022) "On prevention of the spread of tuberculosis in the Russian Federation" and the Order of the Ministry of Health of the Russian Federation of November 15, 2012 No. 932n "On approval of the procedure for providing medical care to tuberculosis patients". Consequently, many studies of the mechanisms of Mtb survival in the lungs of TB patients are necessary to adjust the regulations for the treatment of TB patients and make changes to the Orders of the Ministry of Health of the Russian Federation. To change or correct medical cure, it is necessary to conduct a lot of studies on the relationship between Mtb and host cells in human lungs with the identification of both the mechanisms used by the pathogen for survival bactericidal anti-TB drugs and biomarkers that are able to identify of the pathogen with different growth and phenotypic features. For us, this study presented some part of our work in this direction and it will continue further. In this manuscript we did not want to discuss the results in more detail from a medical point of view, since it is necessary to obtain more data for potential clinical applications of these findings.

Comment 4:

“In the abstract, consider rephrasing "emergence of drug-resistant strains of M. tuberculosis (Mtb)" for clarity.”

Answer for Comment 4:

Now, in Abstract, a passage in line 14 reads:

“The rapid spread of drug-resistant M. tuberculosis (Mtb) strains and phenomenon of …”

Comment 5:

“Throughout the paper, ensure consistency in formatting references and maintain adherence to the journal's guidelines.”

Answer for Comment 5:

I have rechecked all the References and formatted them according to the journal's guidelines.

Comment 6:

“Discussion: add data about predictors factor of adverse event (see and cite  Predictors for Pulmonary Tuberculosis Outcome and Adverse Events in an Italian Referral Hospital: A Nine-Year Retrospective Study (2013-2021). Ann Glob Health. 2022 Apr 26;88(1):26. doi: 10.5334/aogh.3677).”

Answer for Comment 6:

Thank the Reviewer for article with very interesting study about predictor factors of adverse TB event in Bari, Italy. In our work, according to the data from medical records, all the TB patients studied were good citizens of the Russian Federation, permanently living and working in Ural, Siberia, and Yakutia provinces that are characterized by cold climate with a long and very cold winter and a short cool summer. In my opinion, it is these factors that make the main contribution to pure control of TB disease and Mtb infection in these regions. In this manuscript, we did not discuss these aspects, since the main purpose and results of our study did not concern this interesting problem. This problem requires additional research and a broader discussion in future.  

Once again, I would like to thank the Reviewer for careful reading of our manuscript and for their comments that led to improving of article’s contents.

Elena Ufimtseva,             

PhD,

Research Institute of Biochemistry, Federal Research Center of Fundamental and Translational Medicine, Novosibirsk, Siberia, Russia.
